EMBO
Molecular Medicine

# EGFR is required for FOS-dependent bone tumor development via RSK2/CREB signaling

Markus Linder[1], Elisabeth Glitzner[1], Sriram Srivatsa[1], Latifa Bakiri[2], Kazuhiko Matsuoka[2], Parastoo Shahrouzi[1], Monika Dumanic[3], Philipp Novoszel[1], Thomas Mohr[1], Oliver Langer[3,4,5], Thomas Wanek[5], Markus Mitterhauser[3,6], Erwin F Wagner[2] & Maria Sibilia[1,*] (iD)

## Abstract

Osteosarcoma (OS) is a rare tumor of the bone occurring mainly in young adults accounting for 5% of all childhood cancers. Because of the limited therapeutic options, there has been no survival improvement for OS patients in the past 40 years. The epidermal growth factor receptor (EGFR) is highly expressed in OS; however, its clinical relevance is unclear. Here, we employed an autochthonous c-Fos-dependent OS mouse model (H2-*c-fos*LTR) and human OS tumor biopsies for preclinical studies aimed at identifying novel biomarkers and therapeutic benefits of anti-EGFR therapies. We show that EGFR deletion/inhibition results in reduced tumor formation in H2-*c-fos*LTR mice by directly inhibiting the proliferation of cancer-initiating osteoblastic cells by a mechanism involving RSK2/CREB-dependent c-Fos expression. Furthermore, OS patients with co-expression of EGFR and c-Fos exhibit reduced overall survival. Preclinical studies using human OS xenografts revealed that only tumors expressing both EGFR and c-Fos responded to anti-EGFR therapy demonstrating that c-Fos can be considered as a novel biomarker predicting response to anti-EGFR treatment in OS patients.

**Keywords** c-Fos; CREB; epidermal growth factor receptor; osteosarcoma; RSK2
**Subject Categories** Cancer; Musculoskeletal System

## Introduction

Osteosarcoma (OS) is the most common primary tumor of the bone originating from transformed mesenchymal cells (Mirabello *et al*, 2009). It predominantly occurs in young adults with about 400 new cases diagnosed in the United States each year (Ottaviani & Jaffe,

2009). Due to the lack of major therapeutic progress, the 5-year survival of OS patients did not significantly improve since the 1980s, thus raising the need for novel treatment strategies (Ando *et al*, 2013).

To design better therapeutic options for OS patients, it is essential to understand the molecular pathogenesis of the disease. Mutations of the tumor suppressor genes p53 and Rb are major known events inducing osteosarcomagenesis (Marina *et al*, 2004). Moreover, overexpression of proto-oncogenes like c-Myc or c-Fos was reported to play an essential role during bone tumor formation (Wu *et al*, 1990; Gamberi *et al*, 1998). Transgenic mice expressing *c-fos* under the control of the ubiquitously expressed *H2kb* promoter (H2-*c-fos*LTR) develop bone tumors with 100% penetrance (Ruther *et al*, 1989; Grigoriadis *et al*, 1993). Previous studies reported that OS formation in these mice depends on the expression of endogenous *c-fos* and on the activation and stabilization of c-Fos protein via the MAPK-regulated S6 kinase RSK2 (Wang *et al*, 1995; David *et al*, 2005).

Epidermal growth factor receptor (EGFR) expression occurs in 55–81% of all OS cases; however, its clinical relevance in OS development is still controversial (Oda *et al*, 1995; Wen *et al*, 2007; Do *et al*, 2009; Wang *et al*, 2018). EGFR can control c-Fos via the MAPK pathway, and changes in *c-fos* expression after anti-EGFR treatment in cell lines have been suggested to predict therapy success (Jimeno *et al*, 2006; Murphy & Blenis, 2006). Interestingly, EGFR significantly correlates with ERK activation in human OSs (Do *et al*, 2009). Moreover, we recently showed that EGFR is essential in osteoblasts for bone development via ERK-dependent regulation of IGF-1/mTOR signaling (Linder *et al*, 2018). Because of the relatively low incidence rate of OS, clinical trials with inhibitors or antibodies against EGFR could not establish a clear therapeutic benefit in bone tumor formation, due to low statistical power (for review, see Wang *et al*, 2014). Preclinical studies with EGFR inhibitors, which were mainly performed *in vitro* using human OS cell lines or immunodeficient xenograft models, have reported contradictory results (Messerschmitt *et al*,

1  Department of Medicine I, Comprehensive Cancer Center, Institute of Cancer Research, Medical University of Vienna, Vienna, Austria
2  Spanish National Cancer Research Center (CNIO), Madrid, Spain
3  Division of Nuclear Medicine, Department of Biomedical Imaging and Image-Guided Therapy, Medical University of Vienna, Vienna, Austria
4  Department of Clinical Pharmacology, Medical University of Vienna, Vienna, Austria
5  Center for Health & Bioresources, AIT Austrian Institute of Technology GmbH, Seibersdorf, Austria
6  LBI Applied Diagnostics, Vienna, Austria
   *Corresponding author. Tel: +43-1-40160-57502; Fax: 43-1-40160-957502; E-mail: Sibilia-Office@meduniwien.ac.at

2008; Lee *et al*, 2012; Sevelda *et al*, 2015; Gvozdenovic *et al*, 2017).

In the present study, we aimed to understand the role of EGFR during OS development using the autochthonous H2-*c-fos*LTR mouse model and human OS tumor biopsies. Since EGFR can activate S6 and RSK2, we hypothesized that EGFR might act as the upstream receptor inducing OS formation in H2-*c-fos*LTR mice. We found that EGFR signaling is essential in osteoblasts for bone tumor formation and progression. We also identified EGFR as the upstream receptor regulating endogenous *c-fos* expression and stabilizing and activating c-Fos protein via MAPK/CREB signaling. In addition, we show that EGFR and c-Fos co-expression inversely correlate with OS patient survival. These findings provide strong evidence that EGFR and c-Fos expression levels could be used as powerful clinical biomarkers to identify OS patients likely to benefit from anti-EGFR therapy.

# Results

### EGFR is essential for c-Fos-dependent OS formation

In order to investigate the role of EGFR signaling in OS development, we first analyzed *Egfr* and *c-fos* mRNA expression in normal bones from wild-type (wt) and H2-*c-fos*LTR mice, as well as in OSs from H2-*c-fos*LTR mice. Both *Egfr* and *c-fos* levels were significantly elevated in OSs, whereas no genotype depending differences were observed in normal bones (Fig EV1A and B). To assess whether EGFR signaling is required for c-Fos-dependent bone tumors, we genetically attenuated EGFR signaling by breeding H2-*c-fos*LTR mice to mice carrying the hypomorphic *Egfr waved-2* allele ($Egfr^{wa2/wa2}$), leading to reduced EGFR activation (Luetteke *et al*, 1994). H2-*c-fos*LTR/$Egfr^{wa2/wa2}$ mice developed significantly less and smaller bone tumors, indicating that EGFR is essential for bone tumor initiation (Fig 1A–C). Furthermore, serum levels of alkaline phosphatase (ALP), a clinically used indirect OS biomarker (Ren *et al*, 2015a), were also significantly reduced in H2-*c-fos*LTR/$Egfr^{wa2/wa2}$ mice (Fig 1D). To investigate whether pharmacological EGFR inhibition would also show therapeutic benefits, we treated 2-month-old H2-*c-fos*LTR/$Egfr^{wt}$ mice with visible OS, over a period of 5 months with erlotinib (Fig EV1C). Inhibition of EGFR phosphorylation by erlotinib (Fig EV1D) induced tumor stasis indicated by significantly reduced tumor number, tumor size, and ALP serum levels, when compared to vehicle-treated controls (Fig 1E–H). mRNA expression analysis of OS from H2-*c-fos*LTR mice and from human OS from a publicly available dataset [E-GEOD-39058 (Kelly *et al*, 2013)] revealed that EGFR might get activated by an autocrine loop as EGFR ligands were expressed both in mouse and human OS with *Hb-EGF* being the highest (Fig EV1E and F). In line, OSs from H2-*c-fos*LTR mice showed significantly elevated expression of *Hb-EGF* and *TGFa* as compared to normal bones (Fig EV1G and H).

To investigate whether erlotinib showed effective penetration into OSs, 6-month-old H2-*c-fos*LTR/$Egfr^{wt}$ mice with OS, as confirmed with Na[$^{18}$F]F PET imaging, underwent PET scans after i.v. injection of [$^{11}$C]erlotinib at a dose of 2 mg/kg, which corresponds to a human therapeutic dose. Previous results indicated negligible metabolism of [$^{11}$C]erlotinib in mice over the duration of a PET scan, so that radioactivity signal in OSs of H2-*c-fos*LTR/$Egfr^{wt}$

mice can be expected to be only composed of unmetabolized [$^{11}$C]erlotinib, thereby allowing for quantification of erlotinib concentrations in tumor tissue (Traxl *et al*, 2015). PET images revealed that erlotinib penetrated into OSs, while it was excluded from the brain as a pharmacological sanctuary site due to efflux transport at the blood–brain barrier (Fig 1I). Maximum erlotinib concentrations in OSs of individual animals ranged from 3.9 to 10.2 µM, which is in a similar range as peak plasma concentrations of erlotinib achieved in cancer patients after oral dosing (Fig 1J; Katsuya *et al*, 2015).

### EGFR in osteoblasts promotes OS formation in H2-*c-fos*LTR mice

Next, we analyzed the cell type in which EGFR is required to induce OS formation, as we and others have recently shown that EGFR expression in myeloid cells is essential for the development of hepatocellular and colorectal cancer (Lanaya *et al*, 2014; Hardbower *et al*, 2017; Srivatsa *et al*, 2017). We employed the myeloid-specific *LysM-Cre* line and the osteoblast-specific *Runx2-Cre* line to conditionally delete EGFR in myeloid cells or osteoblasts of H2-*c-fos*LTR mice, respectively. EGFR deletion in myeloid cells did not affect OS development in terms of differences in tumor number, burden or serum ALP levels, suggesting that EGFR expression in myeloid cells, which include bone-resorbing osteoclasts, is dispensable during c-Fos-induced OS formation (Appendix Fig S1A–C).

In contrast, deletion of EGFR in the osteoblastic lineage ($Egfr^{\Delta Ob}$) led to significantly reduced tumor number, size, and lower serum ALP levels, demonstrating that EGFR expression in osteoblasts is essential for OS formation (Fig 2A–D). Imaging by µPET/CT with the PET tracer Na[$^{18}$F]F, which is both oncotropic and osteotropic, provides a functional readout on osteoblastic cell activity and tumor burden (Broadhead *et al*, 2015; Rohren *et al*, 2015). This analysis revealed that H2-*c-fos*LTR/$Egfr^{\Delta Ob}$ mice showed significantly reduced standardized uptake values (SUV) of Na[$^{18}$F]F in pelvic OSs at 4 and 7 months of age (Fig 2E and F) emphasizing the requirement for EGFR in osteoblasts for OS development.

### EGFR deletion in osteoblasts decreases proliferation and survival with reduced pRSK2/pCREB/c-Fos activation

As *Egfr* deletion in the osteoblastic lineage affects normal bone development (Linder *et al*, 2018), we next investigated whether EGFR deletion also induces histomorphological changes in c-Fos-dependent OS. Bone volume analysis confirmed that osteoblast-specific deletion of EGFR does not affect the mineralization capacity of OSs (Fig EV2A). Tartrate-resistant acid phosphatase (TRAP) staining further revealed that EGFR is dispensable for osteoclastogenesis in H2-*c-fos*LTR-derived bone tumors (Fig EV2B). Moreover, expression levels of osteoblast-specific differentiation markers were not significantly changed in H2-*c-fos*LTR/$Egfr^{\Delta Ob}$ tumors, indicating that EGFR expression from osteoblastic cells does not affect osteoblast cellularity or differentiation in OSs (Fig EV2C and D).

We next assessed the molecular mechanisms by which osteoblast-specific EGFR deletion impairs OS development. In line with results from the µPET/CT analysis (Fig 2E and F), H2-*c-fos*LTR/$Egfr^{\Delta Ob}$-derived OSs exhibit significantly reduced proliferation and survival compared to H2-*c-fos*LTR/$Egfr^{wt}$ mice, as shown by decreased PCNA and elevated cleaved caspase-3 IHC staining

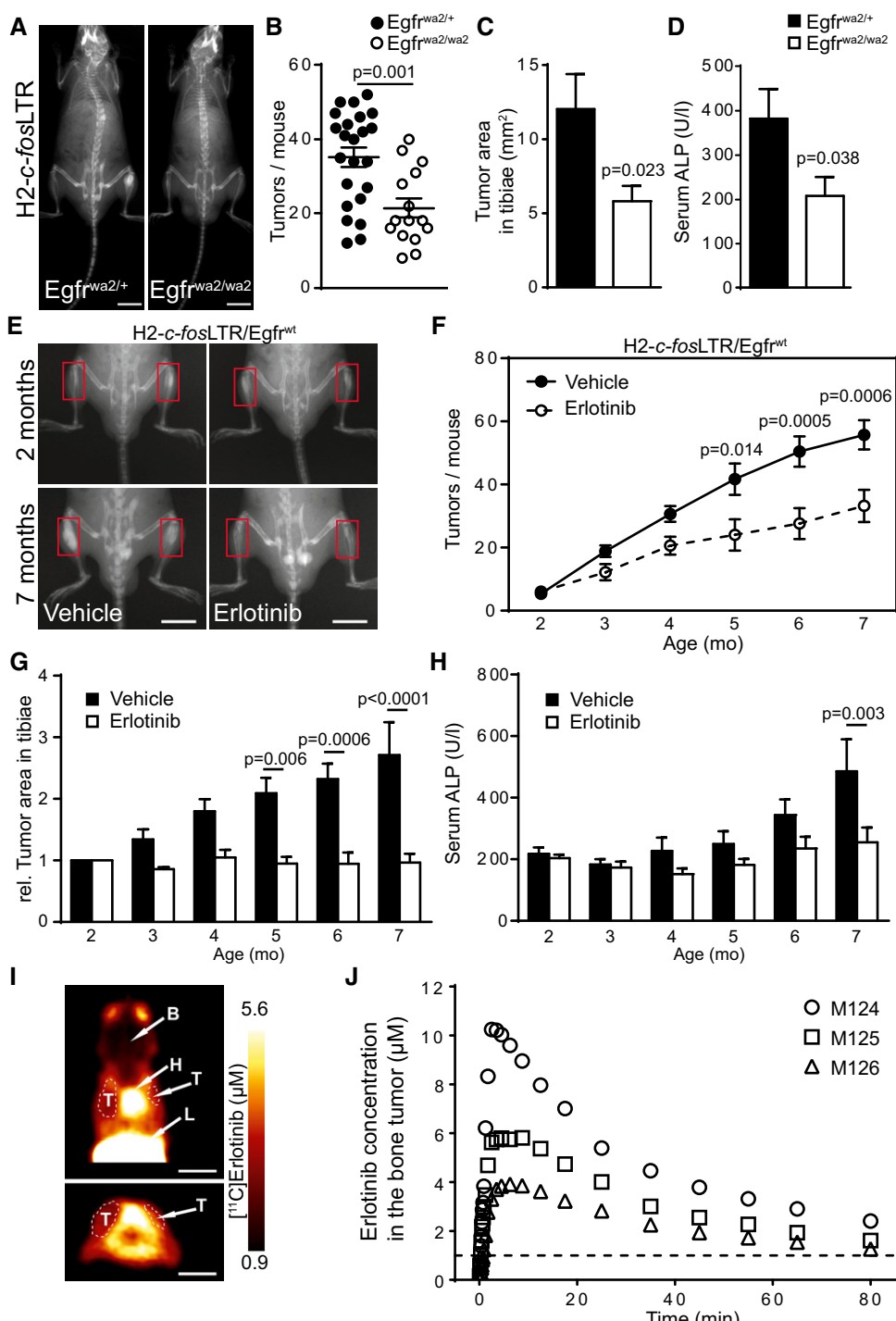

**Figure 1.**

(Fig 3A). In line, OSs from H2-*c-fos*LTR/*Egfr*[ΔOb] mice also displayed decreased expression of cyclin D1 mRNA (Fig 3B) and protein (Fig EV2E). Interestingly, OSs from H2-*c-fos*LTR/*Egfr*[ΔOb] mice had significantly reduced mRNA expression levels of *c-fos,* while transgenic *c-fos* (*c-fos*[tg]) was unaffected indicating that EGFR controls endogenous *c-fos* expression (Fig 3B). Osteoblast-specific EGFR deletion also led to reduced pRSK2, decreased pCREB and less c-

Fos-positive cells in OS, as shown by IHC (Fig 3C) and Western blot analysis of bone tumors from 6- to 7-month-old mice (Fig EV2F). Moreover, EGFR deletion also significantly reduced RSK2-mediated phosphorylation of c-Fos at Ser362 (Fig EV2G). These data indicate that EGFR signaling in osteoblasts is not only essential for proliferation and survival, but also controls endogenous *c-fos* expression and pRSK2/pCREB/c-Fos protein stabilization in c-Fos-dependent OSs.

**Figure 1.  EGFR signaling is essential for c-Fos-dependent bone tumor formation.**

A    X-ray analysis of 6-month-old H2-*c-fos*LTR/Egfr^wa/+ and H2-*c-fos*LTR/Egfr^wa/wa littermates. Scale bars: 1 cm.
B    Bone tumor number per mouse at 5–6 months of age ($n$ = 23 wa/+, 15 wa/wa).
C    Quantification of tumor size in tibiae at 5–6 months of age ($n$ = 23 wa/+, 15 wa/wa).
D    Alkaline phosphatase (ALP) levels in the serum at endpoint (age = 5–6 months; $n$ = 16 wa/+, 11 wa/wa).
E    X-ray analysis before (age = 2 months) and after (age = 7 months) vehicle or erlotinib treatment. Scale bars: 1 cm.
F    Quantification of tumor number during treatment ($n$ = 6 vehicle, 5 erlotinib).
G    Tumor size during treatment ($n$ = 6 vehicle, 5 erlotinib).
H    Analysis of serum ALP levels during treatment ($n$ = 5).
I    PET summation images (0–90 min) in horizontal (upper panel) and coronal view (lower panel) depicting [$^{11}$C]erlotinib distribution in one H2-*c-fos*LTR/*Egfr*^wt mouse (M125). Anatomical structures are labeled with arrows (T, tumor; L, liver; H, heart; B, brain). Scale bars: 1 cm.
J    Concentration–time curves of [$^{11}$C]erlotinib in bone tumors in right scapula of three H2-*c-fos*LTR/*Egfr*^wt mice measured with PET. Broken line indicates threshold for *in vitro* effect of erlotinib (1 μM).

Data information: Data are shown as mean ± SEM. *P*-values were calculated by unpaired, two-tailed *t*-test (B–D) or by two-way ANOVA followed by Bonferroni multiple comparison test (F–H).

## EGFR activates pCREB/c-Fos via MAPK

To further demonstrate the cell-autonomous requirement for EGFR/pRSK2/pCREB/c-Fos activation in osteoblasts, we next analyzed OS cells isolated from H2-*c-fos*LTR/*Egfr*^ΔOb and H2-*c-fos*LTR/*Egfr*^wt mice. In line with our *in vivo* findings, we detected strongly reduced protein levels of pCREB and c-Fos in H2-*c-fos*LTR/*Egfr*^ΔOb OS cells (Fig 4A) and significantly reduced expression of *c-fos* mRNA, while levels of the transgenic *c-fos* (*c-fos*^tg) in OS cells lacking EGFR were comparable to *Egfr*^wt OS cells (Fig 4B). H2-*c-fos*LTR/*Egfr*^ΔOb OS cells also showed significantly increased apoptosis and reduced

proliferation as evidenced by elevated levels of cleaved caspase-3 and cleaved PARP (Fig EV3A) and reduced Ki-67-positive cells (Fig EV3B) when compared to H2-*c-fos*LTR/*Egfr*^wt OS cells.

Moreover, EGFR inhibition with erlotinib resulted in a dose-dependent downregulation of pRSK2/pCREB/c-Fos in H2-*c-fos*LTR/*Egfr*^wt OS cells (Fig 4C) leading to reduced proliferation as shown by reduced cyclin D1 protein levels (Fig EV3C) and less Ki-67-positive cells (Fig EV3D). Erlotinib treatment also led to significantly reduced *c-fos* mRNA expression without affecting transgenic *c-fos* mRNA levels, further demonstrating that EGFR signaling specifically promotes transcription of endogenous *c-fos* (Fig 4D).

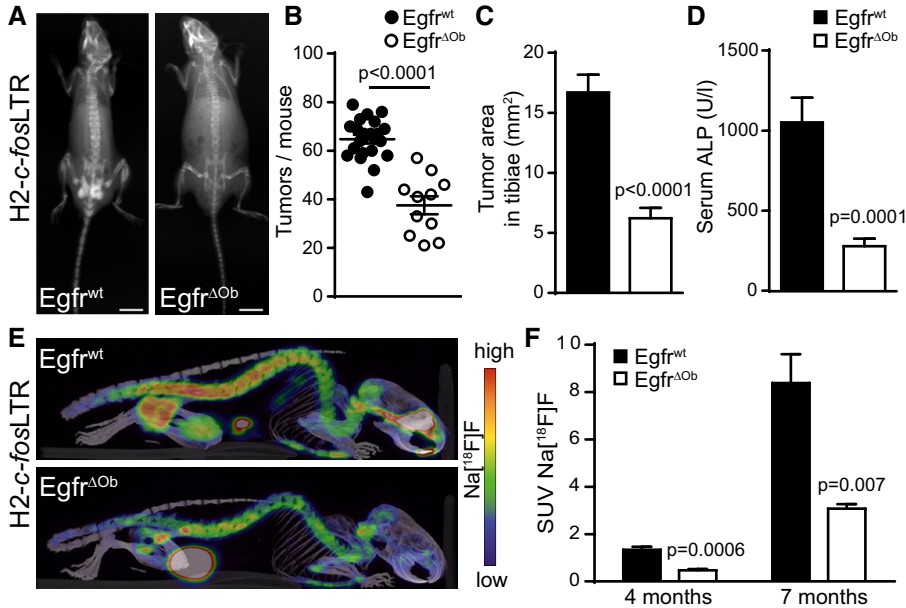

**Figure 2.  Osteoblast-specific EGFR deletion reduces c-Fos-driven OS formation.**

A    X-ray analysis of 7-month-old H2-*c-fos*LTR/Egfr^wt and H2-*c-fos*LTR/Egfr^ΔOb littermates. Scale bars: 1 cm.
B    Bone tumor number per mouse at 6–7 months of age ($n$ = 22 wt, 11 ΔOb).
C    Quantification of tumor size in tibiae ($n$ = 22 wt, 11 ΔOb).
D    ALP levels in the serum at 7-month endpoint ($n$ = 17 wt, 8 ΔOb).
E    μPET/CT analysis of 7-month-old H2-*c-fos*LTR/Egfr^wt and H2-*c-fos*LTR/Egfr^ΔOb littermates.
F    Standardized uptake values (SUV) of the μPET tracer Na[$^{18}$F]F in the pelvic OS of 4- and 7-month-old H2-*c-fos*LTR/Egfr^wt and H2-*c-fos*LTR/Egfr^ΔOb mice. $n$ = 6 wt, 4 ΔOb for 4-month time-point, $n$ = 6 wt, 3 ΔOb for 7-month time-point.

Data information: Data are shown as mean ± SEM. *P*-values were calculated by unpaired, two-tailed *t*-test.

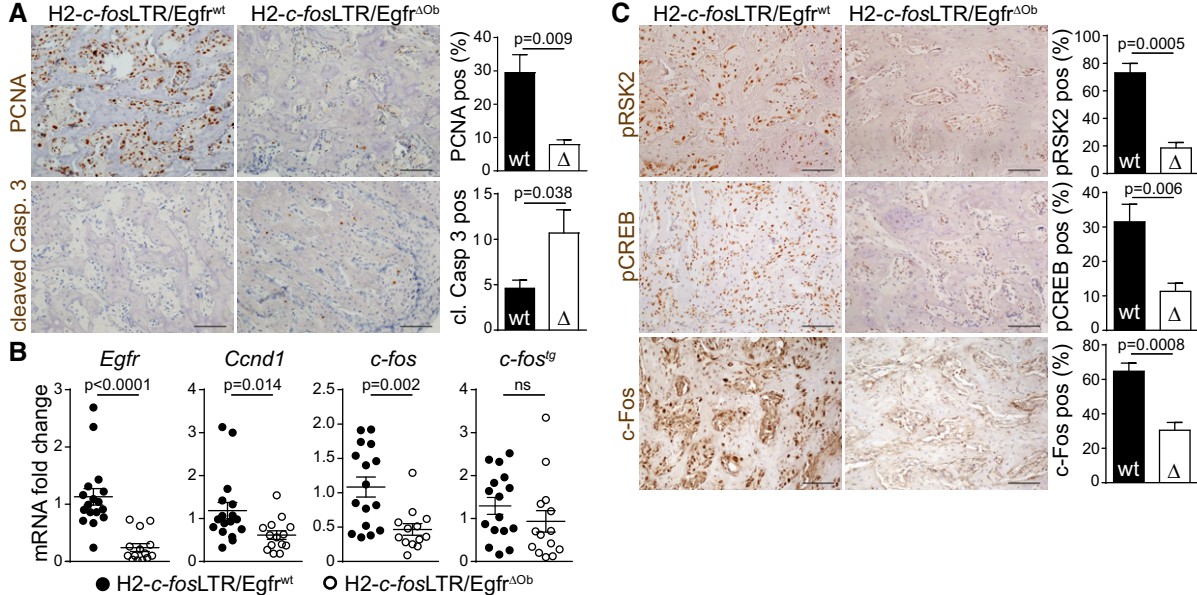

**Figure 3.  EGFR is essential for proliferation, survival and c-Fos protein and mRNA expression via RSK2/CREB phosphorylation.**

A   PCNA and cleaved caspase-3 IHC staining and quantification shown as % positive cells (for PCNA) and as positive cells per mm² (for cleaved caspase-3) in OS from H2-*c-fos*LTR/Egfr^wt (*n* = 6) and H2-*c-fos*LTR/Egfr^ΔOb (*n* = 5) mice. Scale bars: 100 μm.

B   *Egfr*, *Ccnd1*, *c-fos* and *transgenic c-fos* (*c-fos*^tg) mRNA expression levels in tumors of H2-*c-fos*LTR/Egfr^wt and H2-*c-fos*LTR/Egfr^ΔOb mice normalized to *Tbp*. *n* = 17 wt, 14 ΔOb (*Egfr*, *Ccnd1*), *n* = 16 wt, 13 ΔOb (*c-fos*), *n* = 16 wt, 14 ΔOb (*c-fos*^tg).

C   IHC staining and quantification showing pRSK2- (*n* = 4), pCREB- (*n* = 7 wt, 6 ΔOb), and c-Fos (*n* = 5)-positive cells (%) in OS from H2-*c-fos*LTR/Egfr^wt and H2-*c-fos*LTR/Egfr^ΔOb littermates. Scale bars: 100 μm.

Data information: Data are shown as mean ± SEM. *P*-values were calculated by unpaired, two-tailed *t*-test.

We next dissected the molecular signaling pathway downstream of EGFR responsible for activation of pCREB and c-Fos. Because RSK2 activation depends on ERK1/2 and PI3K-dependent PDK-1 signaling (Anjum & Blenis, 2008), serum-starved H2-*c-fos*LTR/*Egfr*^wt OS cells were pre-treated with inhibitors against EGFR (afatinib), PDK1 (GSK2334470), mTOR (rapamycin), or ERK1/2 (U0126) and afterwards stimulated with EGF (50 ng/ml) for 0, 5, or 30 min (Fig 4E and F). As expected, EGF strongly induced RSK2/CREB phosphorylation after 5 min followed by elevated c-Fos protein levels after 30 min of stimulation (Fig 4E). Comparable expression patterns were also detected in cells pre-treated with a PDK-1 or mTOR inhibitor, suggesting that the PI3K pathway is dispensable for pCREB/c-Fos activation (Fig 4E). Importantly, when ERK1/2 was blocked, upregulation of the two target proteins was not detected (Fig 4E), indicating that EGF-induced activation of pCREB/c-Fos in *c-fos* transgenic OS cells depends on MAPK signaling.

To further demonstrate that EGFR signaling is mainly affecting the expression of endogenous but not transgenic *c-fos*, primary OS cells isolated from *p53*^f/f *Rb1*^f/f *Osx*-Cre mice were employed. It has been reported, that *c-fos* is upregulated in primary OS cells isolated from these mice (Walkley *et al*, 2008). In agreement with the results obtained with primary H2-*c-fos*LTR OS cells, also *p53*^f/f *Rb1*^f/f *Osx*-Cre OS cells showed reduced pRSK2/pCREB/Fos protein levels (Fig 4G) and significantly downregulated *c-fos* mRNA expression after erlotinib treatment (Fig 4H). On the other hand, strong activation of pEGFR/pRSK2/pCREB/c-Fos was induced after EGF stimulation (Fig 4I). These data indicate that the observed mechanism of

EGFR-dependent activation of c-Fos via pRSK2/pCREB also applies for bone tumors that are not induced by transgenic *c-fos*.

### Osteoblast-specific overexpression of the EGFR ligand AREG accelerates OS formation

Next, we wanted to investigate whether constitutive activation of the EGFR pathway can enhance bone tumor formation. We therefore bred H2-*c-fos*LTR mice to mice overexpressing the EGFR ligand Amphiregulin (AREG) under the control of the osteoblast-specific *col1a1* promoter (*ColAREG*) (Vaidya *et al*, 2015). Double transgenic mice (H2-*c-fos*LTR/*ColAREG*) showed significantly increased tumor numbers, size and a tendency of elevated serum ALP levels (Fig 5A–D), indicating that autocrine osteoblast-specific EGFR activation accelerates OS development.

Tumors from double transgenic mice displayed significantly increased AREG protein (Appendix Fig S2A) and mRNA expression levels (Appendix Fig S2B) and elevated EGFR phosphorylation (Appendix Fig S2C). Moreover, they showed significantly increased *c-fos* mRNA expression levels (Fig 5E), whereas the transgenic *c-fos* mRNA in the tumor was not significantly changed (Appendix Fig S2D). Western blot analysis further revealed that bone tumors of H2-*c-fos*LTR/*ColAREG* mice displayed elevated activation of the EGFR downstream proteins pRSK2, pCREB and c-Fos resulting in elevated proliferation with increased cyclin D1 protein expression (Fig 5F) and decreased apoptosis as shown by reduced cleaved caspase-3 levels (Fig 5G). These data demonstrate that constitutive AREG-induced EGFR activation accelerates tumorigenesis in H2-*c-fos*LTR

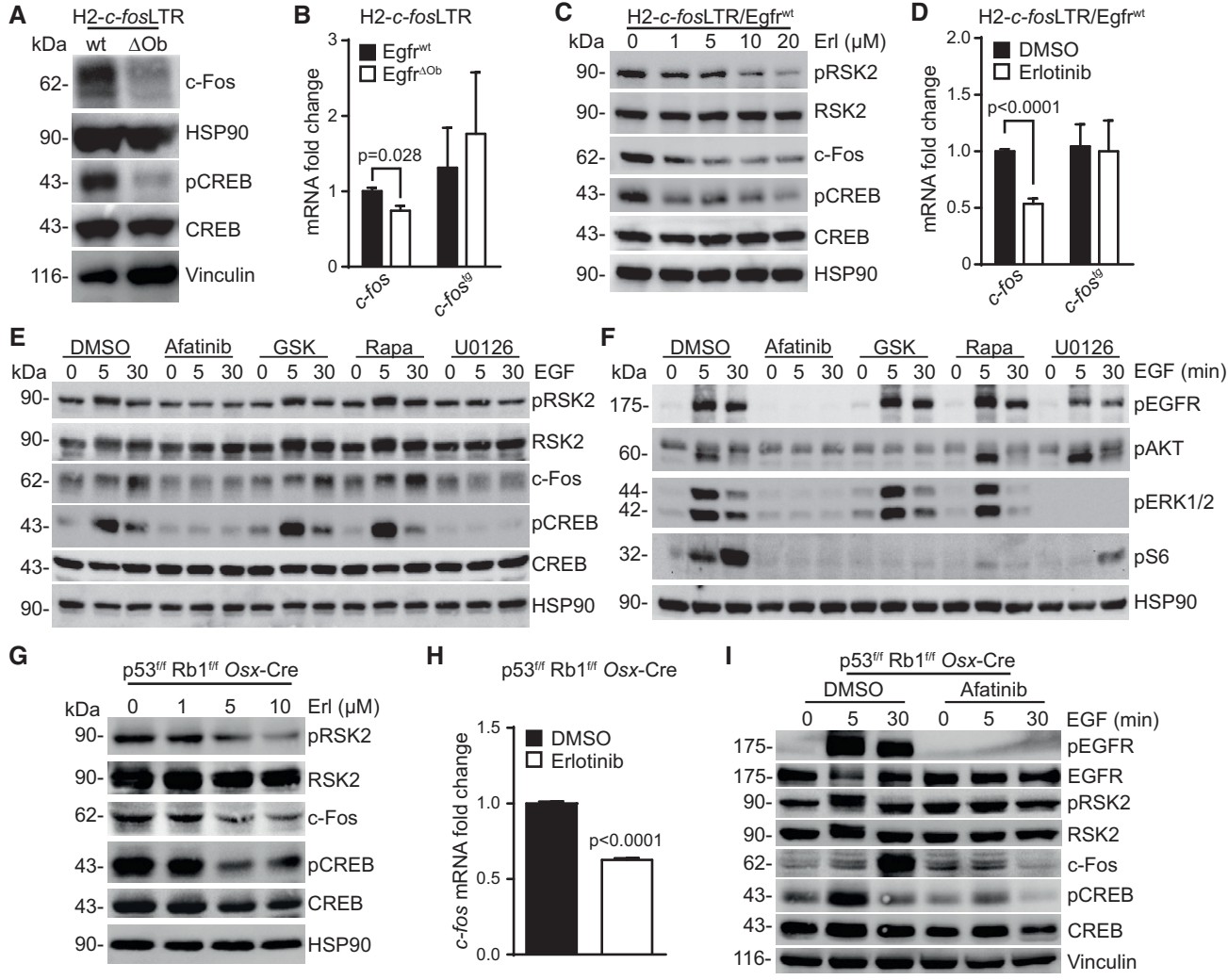

**Figure 4.  EGFR controls c-Fos via MAPK-dependent CREB activation in primary OS cells.**

A    Western blot analysis of primary OS cells isolated from H2-*c-fos*LTR/Egfr$^{wt}$ and H2-*c-fos*LTR/Egfr$^{\Delta Ob}$ mice.

B    *c-fos* and *transgenic c-fos* (*c fos$^{tg}$*) mRNA expression levels in primary H2-*c-fos*LTR OS cells after 4× *in vitro* passages, cultured under standard conditions (*n* = 3 independent cell lines).

C    Western blot analysis of H2-*c-fos*LTR/Egfr$^{wt}$ OS cells treated for 24 h with erlotinib.

D    *c-fos* and *c-fos$^{tg}$* mRNA expression levels in H2-*c-fos*LTR/Egfr$^{wt}$ OS cells treated for 24 h with erlotinib (10 μM) or DMSO as control (*n* = 4 independent cell lines).

E, F  Western blot analysis of starved H2-*c-fos*LTR/Egfr$^{wt}$ OS cells, pre-treated with DMSO (1:1,000), afatinib (5 μM), GSK2233470 (10 μM), rapamycin (10 nM), or U0126 (10 μM) for 30 min and stimulated with EGF (50 ng/ml) as indicated.

G    Western blot analysis of primary OS cells isolated from a *p53$^{f/f}$ Rb1$^{f/f}$ Osx*-Cre mouse after 24-h erlotinib treatment.

H    c-fos mRNA expression levels in *p53$^{f/f}$ Rb1$^{f/f}$ Osx*-Cre OS cells treated for 24 h with erlotinib (10 μM; *n* = 3).

I    Western blot analysis of starved *p53$^{f/f}$ Rb1$^{f/f}$ Osx*-Cre OS cells, pre-treated with DMSO (1:1,000) or afatinib (5 μM) for 30 min and stimulated with EGF (50 ng/ml) as indicated.

Data information: Data are shown as mean ± SEM. *P*-values were calculated by unpaired, two-tailed *t*-test.
Source data are available online for this figure.

mice via enhanced tumor cell proliferation and survival resulting from increased pRSK2/pCREB/c-Fos signaling.

### EGFR and c-Fos co-expression correlates with poor survival of human OS patients

To investigate whether our findings in the FOS-driven OS mouse model are of clinical relevance, we stained biopsies from 52

human OS patients with antibodies directed against EGFR and c-Fos and divided them, according to their expression levels, into two groups (positive or negative; Fig 6A). In line with published studies (Do *et al*, 2009; Lee *et al*, 2012), we did not find any significant correlation between EGFR expression and overall survival of OS patients (Fig EV4A). Also c-Fos expression in bone tumors did not show any significant impact on the overall patient survival (Fig EV4B).

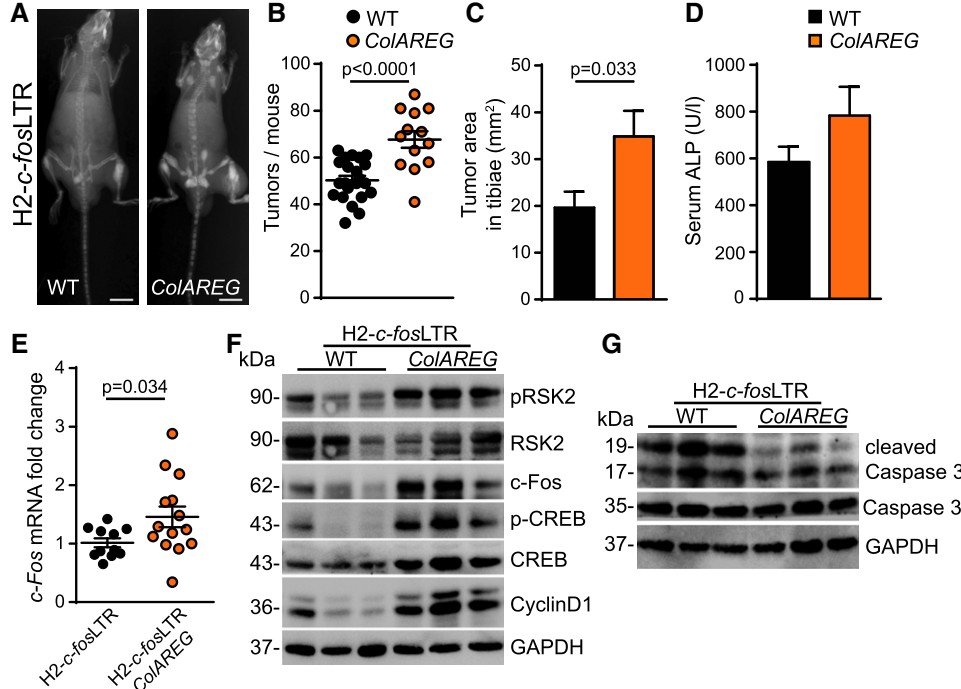

**Figure 5.   Osteoblast-specific overexpression of the EGFR ligand Amphiregulin accelerates tumor formation in H2-*c-fos*LTR mice.**

A   X-ray analysis of 6-month-old H2-*c-fos*LTR and H2-*c-fos*LTR/*ColAREG* littermates. Scale bars: 1 cm.
B   Bone tumor number per mouse at 5–6 months of age (*n* = 22 wt, 13 *ColAREG*).
C   Quantification of tumor size in tibiae at 5–6 months of age (*n* = 22 wt, 13 *ColAREG*).
D   Alkaline phosphatase (ALP) levels in the serum at endpoint (age = 5–6 months; *n* = 29 wt, 19 *ColAREG*).
E   *c-fos* mRNA expression levels in OSs of H2-*c-fos*LTR (*n* = 11) and H2-*c-fos*LTR/*ColAREG* mice (*n* = 14).
F, G   Western blot analysis of bone tumor protein lysates from H2-*c-fos*LTR and H2-*c-fos*LTR/*ColAREG* mice.

Data information: Data are shown as mean ± SEM. *P*-values were calculated by unpaired, two-tailed *t*-test.
Source data are available online for this figure.

However, when we stratified patients according to the co-expression of EGFR and c-Fos we found significantly reduced survival of patients with tumors that showed high levels of both proteins (Fig 6B), suggesting that EGFR inhibition might be a possible treatment option for patients suffering from EGFR and FOS double-positive OS.

Analysis of RNAseq data of human OS from a publicly available dataset [GSE39058 (Kelly *et al*, 2013)] further revealed that *EGFR* expression significantly correlated with *FOS* expression (Fig 6C). In contrast, except for *PDGFRB*, other cancer-associated receptor tyrosine kinases (RTKs) showed either negative or no correlation with *FOS* expression (Fig 6C). However, applying linear models for microarray and RNAseq data (LIMMA) analysis revealed that human OS high for both *EGFR* and *FOS* did not show any significantly changed transcriptional profile as compared to double-negative OS (Dataset EV1). Among EGFR ligands, only *TGFa* showed a significant correlation with *FOS* expression, indicating that *TGFa* might be crucial for EGFR activation in FOS-dependent OSs (Fig EV4C).

### EGFR signaling in human OS cell lines and orthotopic tumors

We next tested our hypothesis that EGFR inhibition is more effective in OSs positive for both EGFR and c-Fos, and analyzed their expression levels in a panel of human OS cell lines. Among six tested cell lines, only the highly aggressive 143b cells showed co-expression of both EGFR and c-Fos protein and mRNA, whereas LM7 cells expressed only detectable levels of EGFR but not of c-Fos (Figs 7A and EV5A). Moreover, EGF stimulation of starved LM7 cells did not induce c-Fos protein expression, excluding the possibility that c-Fos gets activated via EGFR signaling in this cell line (Fig EV5B).

In line with our hypothesis, *in vitro* erlotinib treatment significantly reduced the cell viability in 143b cells, but not in LM7 cells (Fig 7B). Conversely, EGF stimulation led to elevated cell viability only in 143b, but not in LM7 cells (Fig 7C), indicating that OS cells become addicted to EGFR signaling only if they express both EGFR and c-Fos. To demonstrate that c-Fos is essential in 143b OS cells, we knocked down *c-fos* in these cells by employing doxycycline-inducible shRNA (Fig EV5C and D) to show that both cell proliferation and cell viability were reduced in 143b OS cells lacking *c-fos* (Fig EV5E–H).

An orthotopic OS model mimicking the human disease was employed to investigate whether our findings also hold true *in vivo*. For this purpose, the human OS cell lines 143b and LM7 were injected into the proximal marrow space of the tibia bone of immunosuppressed mice. After tumors had developed, mice were divided into groups with equal mean tumor volumes and treated with erlotinib (50 mg/kg/day) or vehicle (0.5% Methylcellulose; Fig 7D and G).

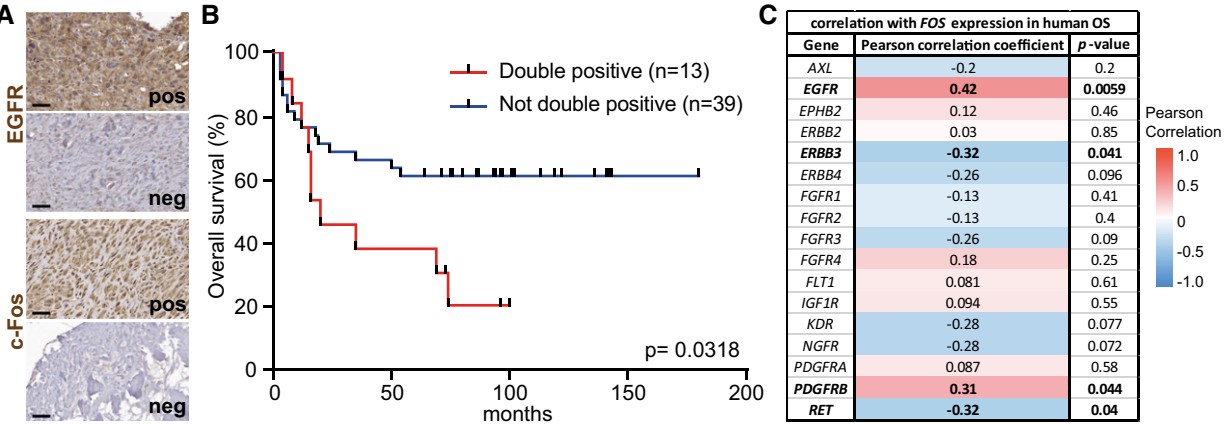

**Figure 6. EGFR and c-Fos co-expression in human OS negatively correlates with patient survival.**

A  Representative images of OS biopsies stained with antibodies against EGFR and c-Fos. Scale bars: 50 μm.
B  Kaplan–Meier survival curve comparing the survival of patients with EGFR and c-Fos double-positive OSs against patients without co-expression of both proteins (*n* = 52).
C  Gene expression correlation analysis of several cancer-associated RTKs with *FOS* in human OS (data analyzed from publicly available dataset E-GEOD-39058).

Data information: *P*-values were calculated by log-rank (Mantel–Cox) test comparing the two Kaplan–Meier curves (B) or by unpaired, two-tailed *t*-test (C).

Similar to what was observed *in vitro*, 143b-derived xenografts showed high EGFR and c-Fos protein expression, whereas LM7-induced tumors only expressed EGFR (Fig EV5I) but no c-Fos (Fig EV5J). Importantly and in agreement with patient survival data (Fig 6B), xenografts from EGFR and c-Fos double-positive 143b cells were more aggressive, resulting in significantly shorter median survival and twofold to threefold increased tumor progression as compared to LM7-derived tumors (Yuan *et al*, 2009; Ren *et al*, 2015b). Erlotinib treatment strongly reduced EGFR phosphorylation in both xenograft models (Fig EV5K). Erlotinib-induced tumor reduction at the endpoint was exclusively detectable in 143b-derived OS, whereas LM7-derived OS did not show any significant differences in tumor size or weight after erlotinib therapy (Fig 7D–I). The results from this preclinical trial demonstrate that EGFR inhibition only reduces OS progression in tumors that co-express both EGFR and c-Fos (Fig 7E and F) confirming EGFR as a driver in this subclass of OS.

We next assessed whether in the 143b OS xenograft model EGFR inhibition also affected proliferation and survival via pCREB/c-Fos as seen in H2-*c-fos*LTR mice. IHC stainings revealed significantly reduced levels of PCNA and increased numbers of cleaved caspase-3-positive cells in tumors after erlotinib treatment (Fig 7J). This, together with reduced protein levels of Ki-67 and cyclin D1 (Fig EV5L and M), indicates that EGFR plays an essential role during proliferation and apoptosis in orthotopically injected 143b OS cells. Additionally, we found significantly reduced pCREB/c-Fos levels in 143b-derived tumors after erlotinib treatment as shown by IHC (Fig 7K) and Western blot analysis (Fig 7L). In summary, these data provide strong evidence that EGFR-dependent activation of pCREB/c-Fos represents a driving pathway in EGFR/Fos double-positive OS.

## Discussion

EGFR inhibition in human OS cell lines has been shown to effectively decrease cell migration, cell invasion and colony formation

(Messerschmitt *et al*, 2008). Other studies suggested that EGFR inhibition has no significant influence on cell proliferation, but enhances the anti-tumor effects of starvation or chemotherapy-induced cell stress *in vitro* (Sevelda *et al*, 2015). Our study shows that EGFR is a driver of OS, at least in a subgroup of patients suffering from EGFR and c-Fos double-positive tumors.

We found significantly reduced OS formation in H2-*c-fos*LTR mice, when we either genetically deleted EGFR or pharmacologically inhibited the receptor. Moreover, cell-type-specific EGFR knock-out models revealed that EGFR signaling in osteoblasts is the driving force essential for c-Fos-dependent OS formation, whereas myeloid-cell-specific deletion of EGFR did not affect osteosarcomagenesis. This is in contrast to our previous findings in colorectal and hepatocellular carcinoma, where EGFR expression in myeloid cells was found to be essential for tumor formation (Lanaya *et al*, 2014; Srivatsa *et al*, 2017), indicating that the tumorigenic role of EGFR varies depending on the tumor and cell type.

As bone tumors in H2-*c-fos*LTR mice arise from the osteoblastic lineage (Grigoriadis *et al*, 1993), we hypothesize that the tumor cell-intrinsic function of EGFR might be similar to its role in pancreatic ductal adenocarcinoma (PDAC), where EGFR deletion or inhibition was shown to ameliorate KRAS-driven PDAC (Ardito *et al*, 2012; Navas *et al*, 2012). It would be of interest to address whether the molecular mechanisms in c-Fos-dependent OSs also hold true for KRAS-mutated PDACs in which c-Fos was also reported to be a major driver (Guo *et al*, 2015; You *et al*, 2016).

Clinical studies have linked expression of the EGFR ligand AREG to poor prognosis of patients suffering from carcinomas such as pancreatic, breast, or colorectal cancer (Willmarth & Ethier, 2008; Yamada *et al*, 2008; Wang *et al*, 2016). A recent study by Liu *et al* (2015) concluded that AREG also promotes cell migration and metastatic capacity of human OS cells as shRNA-mediated knock-down of AREG resulted in significantly less lung metastasis nodules after i.v. injection. In line with this study, we found significantly elevated tumor formation and progression when we boosted EGFR signaling

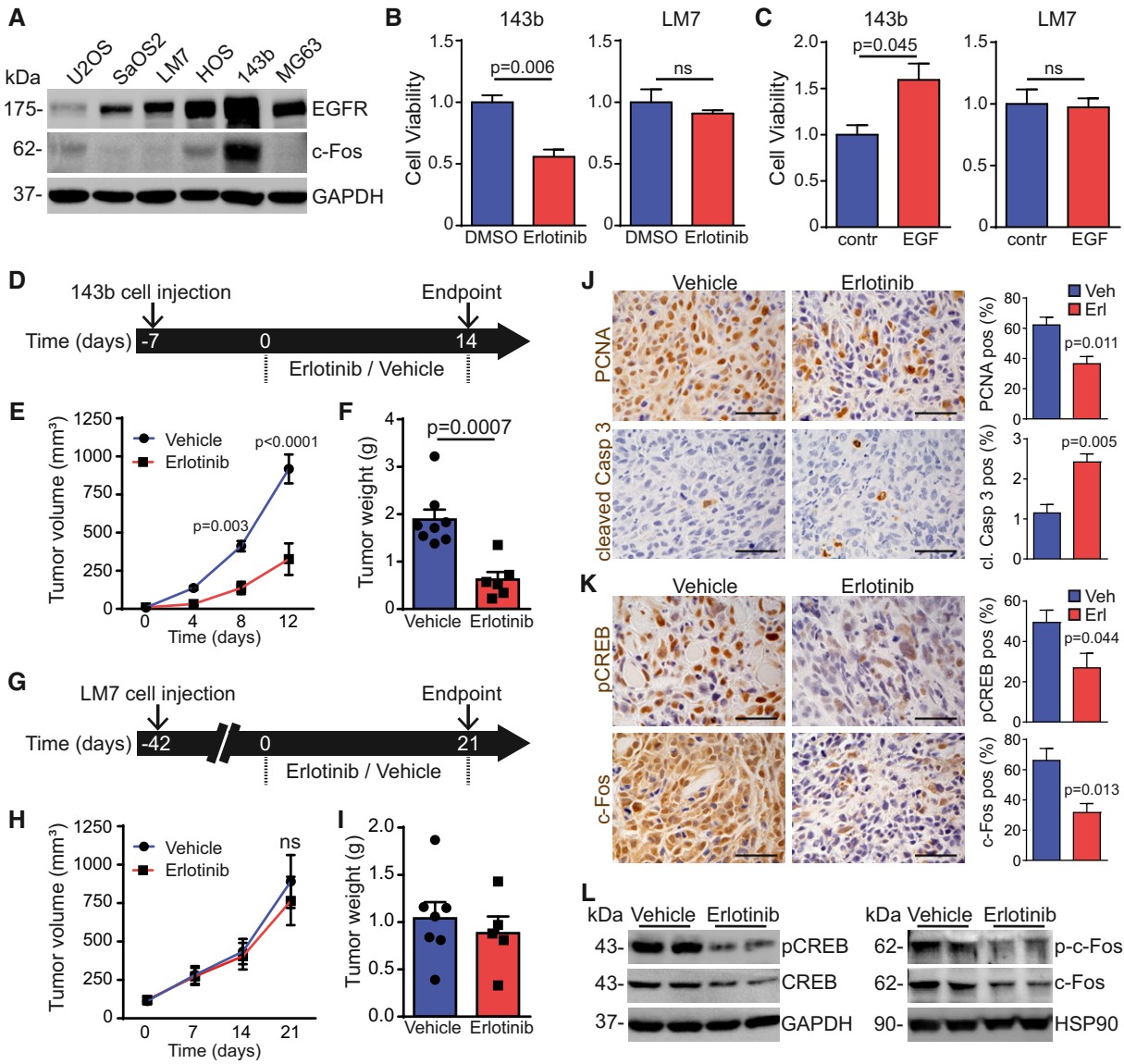

**Figure 7. EGFR inhibition decreases tumor progression of human OS cells 143b after orthotopic injection.**

A   Western blot analysis showing EGFR and c-Fos protein levels in human OS cell lines cultured under standard conditions.

B   Cell viability of 143b or LM7 cells cultured for 24 h in medium (+10% FCS) with DMSO (1:2,000) or erlotinib (10 μM; $n = 3$, representative result from three independent experiments, shown as fold change, normalized to DMSO).

C   Cell viability of 143b or LM7 cells cultured for 24 h in medium (+0.5% FCS) containing 1× PBS (contr., 1:200) or EGF (50 ng/ml; $n = 3$, representative result from three independent experiments, shown as fold change, normalized to control).

D   Treatment scheme: 143b OS cells ($10^6$ in 25 μl PBS) were intratibially injected; mice were treated for 14 days with vehicle or erlotinib (50 mg/kg) 7 days after injection, when the tumors started to grow.

E   143b xenograft growth curve during therapeutic regime ($n = 8$ vehicle, 6 erlotinib; two independent experiments).

F   143b tumor weight at endpoint ($n = 8$ vehicle, 6 erlotinib; two independent experiments).

G   Treatment scheme: LM7 OS cells ($10^6$ in 25 μl PBS) were intratibially injected; mice were treated for 21 days with vehicle or erlotinib (50 mg/kg) 42 days after injection, when the tumors started to grow.

H   LM7 xenograft growth curve during therapeutic regime ($n = 7$ vehicle, 5 erlotinib; two independent experiments).

I   LM7 tumor weight at endpoint ($n = 7$ vehicle, 5 erlotinib, two independent experiments).

J   IHC analysis of PCNA and cleaved caspase-3 in 143b-derived tumors ($n = 4$). Scale bars: 50 μm.

K   IHC analysis of pCREB and c-Fos in 143b-derived tumors ($n = 5$ for pCREB and $n = 4$ for c-Fos analysis). Scale bars: 50 μm.

L   Western blot analysis of pCREB/CREB and p-c-Fos/c-Fos protein expression in lysates directly isolated from 143b xenografts at endpoint, after 14 days of vehicle or erlotinib treatment.

Data information: Data are shown as mean ± SEM. *P*-values were calculated by unpaired, two-tailed *t*-test (B, C, F, I, J, and K) or by two-way ANOVA followed by Bonferroni multiple comparison test (E and H).

Source data are available online for this figure.

in bone tumors by crossing H2-*c-fos*LTR mice to mice that overexpress AREG under an osteoblast-specific promoter (*ColAREG*), suggesting that AREG could indeed be a clinically relevant molecule in OS development.

Mechanistically, we found that EGFR is essential for proliferation, survival, and MAPK-dependent activation of pCREB/c-Fos in bone tumors of H2-*c-fos*LTR mice and in orthotopically injected human 143b OS cells, which are positive for both EGFR and Fos. As it has been reported that the MAPK-signaling protein RSK2 is essential for progression of OSs in H2-*c-fos*LTR mice (David *et al*, 2005), it is likely that EGFR acts as the upstream receptor controlling RSK2 phosphorylation, which in turn activates and stabilizes the c-Fos protein leading to OS progression. In addition, it has been shown that endogenous *c-fos* expression levels directly correlate with tumor formation in H2-*c-fos*LTR mice (Wang *et al*, 1995). Mice heterozygous for *c-fos* ($c\text{-}fos^{+/-}$) developed about 50% less OSs, whereas *c-fos* full knock-out mice ($c\text{-}fos^{-/-}$) were nearly tumor-free (Wang *et al*, 1995). In this study, we discovered that EGFR signaling is essential for endogenous *c-fos* mRNA expression. Interestingly, RSK2 deletion was shown not to affect endogenous *c-fos* transcription in bone tumors of H2-*c-fos*LTR mice (David *et al*, 2005). Therefore, we speculate that EGFR controls *c-fos* transcription directly via ERK1/2, which has indeed been shown to induce *c-fos* mRNA expression after binding to RSK2-independent phosphorylation sites on the carboxyl-terminal transactivation domain (Monje *et al*, 2003). While we can distinguish transgenic *c-fos* mRNA from total *c-fos* mRNA, we cannot distinguish the c-Fos protein expressed from the transgene. Therefore, we can formally not rule out the possibility that EGFR also affects transgenic c-Fos protein levels. To confirm that EGFR mainly plays a role in endogenous c-Fos activation, we used an independent preclinical OS model, which is based on osteoblast-specific deletion of the two tumor suppressor genes p53 and Rb1 (Walkley *et al*, 2008). Also in this model, *c-fos* is upregulated in $p53^{f/f}$ $Rb1^{f/f}$ *Osx*-Cre OS cells suggesting a potential tumorigenic role for c-Fos in this model (Walkley *et al*, 2008). We found that *c-fos* mRNA and pRSK2/pCREB/c-Fos protein levels were significantly reduced after EGFR inhibition, demonstrating that the here proposed mechanism does likely not depend on the *c-fos* transgene.

By analyzing biopsies from human OS patients, we could not detect any significant impact of EGFR expression on overall patient survival, which is similar to previous reports (Do *et al*, 2009; Lee *et al*, 2012). However, histological analysis of OS patients by Wen *et al* (2007) revealed high EGFR expression in 4 out of 5 (80%) recurring lesions, whereas only 14 out of 37 (38%) primary tumors showed such high EGFR protein levels. Surprisingly, one study reported a positive correlation between EGFR expression and good clinical outcome in high-grade OS patients, which might be due to the fact that patients with high EGFR levels show better responses to conventional therapies (Kersting *et al*, 2007). In contrast, another recent report found a significant correlation between high EGFR levels in OS and reduced patient survival (Wang *et al*, 2018). Since OS is a rare tumor, the few numbers of OS patients preclude a comprehensive investigation. Thus, the potential of anti-EGFR agents in the treatment of human OS remained elusive, raising the need to identify prognostic molecular markers for future treatment approaches. When we categorized OS patients according to the co-expression of both proteins, EGFR and c-Fos, we found significantly

reduced overall survival within this sub-population. Thus, staining OS patient biopsies for these two biomarkers could be a valuable approach to predict disease progression. Although c-Fos could be downstream of many other RTKs expressed in OS, RNAseq analysis revealed that besides *EGFR*, only *PDGFRB* significantly correlated with *FOS* expression indicating that PDGFRβ signaling might also act as an upstream activator for c-Fos-dependent OS. Previous reports have shown that PDGFR is indeed frequently overexpressed in OS and preclinical studies further suggested a tumor-promoting role for PDGFR (McGary *et al*, 2002; Kubo *et al*, 2008).

In future, EGFR and c-Fos could be useful biomarkers to stratify OS patients. A previous report showed that *c-fos* expression changed in human squamous cell carcinomas after EGFR-TKI treatment and correlated with therapy response in xenograft models (Jimeno *et al*, 2006). However, the potential use of FOS for predicting therapy response was not demonstrated. In the current study, we show that only OSs which express c-Fos respond to anti-EGFR treatment, thereby identifying FOS as a potent predictive biomarker in OS patients. We found that orthotopic bone tumors induced with the EGFR and c-Fos expressing 143b OS cell line were responsive to anti-EGFR treatment, whereas LM7 cells, which only express EGFR but not c-Fos, did not respond to erlotinib. A recent study also reported that treatment with cetuximab, a clinically used anti-EGFR antibody, led to significantly reduced tumor formation after intratibial injection of 143b cells (Gvozdenovic *et al*, 2017). Taken together, our results indicate that anti-EGFR therapies will be successful in patients with FOS-positive OS, where EGFR seems to be a tumor driver. These results might allow better OS patient stratification and therefore offer new therapeutic approaches for a rare tumor type for which there are basically no treatment options.

## Materials and Methods

### Mice

$Egfr^{f/f}$ (Natarajan *et al*, 2007), $Egfr^{wa2/wa2}$ (Luetteke *et al*, 1994), *Runx2-Cre* (Rauch *et al*, 2010), *LysM-Cre* (Clausen *et al*, 1999), H2-*c-fos*LTR (Ruther *et al*, 1989), and *ColAREG* (Vaidya *et al*, 2015) mice have been described previously. $Egfr^{f/f}$, H2-*c-fos*LTR, *Runx2-Cre*, and *LysM-Cre* mice were maintained on a pure C57/BL6J background. $Egfr^{wa2/wa2}$ and *ColAREG* mice were maintained on a mixed genetic background. Both males and females were used indiscriminately. Mice were analyzed between 2 and 7 months of age as indicated in the corresponding figure legend. The animals were monitored and analyzed without randomization or blinding. Mice were kept in the animal facility of the Medical University of Vienna with unlimited access to both water and standard laboratory chow under a light–dark cycle of 12 h at 22°C in accordance with institutional policies and federal guidelines. All animal experiments conducted were compliant with federal laws and guidelines of the Medical University of Vienna.

### Cell culture

Human OS cell lines U2OS, SaOS2, HOS, 143b, and MG63 were purchased from ATCC (Manassas, VA). The LM-7 OS cell line (Jia *et al*, 1999) was provided by Dr. Eugenie S. Kleinerman from the

M.D. Anderson Cancer Center (University of Texas). 143b OS cells have been reported not to carry any EGFR mutations (Wen *et al*, 2007). 143b cells were cultured in DMEM supplemented with 10% FCS (Sigma), and LM7 cells were cultured in MEM with non-essential amino acids, pyruvate, and 10% FCS (Sigma). Primary mouse OS cells were isolated and cultured as previously described (Grigoriadis *et al*, 1993). For EGFR inhibitor experiments, sub-confluent cells were treated with indicated concentrations of erlotinib (Santa Cruz Biotechnology) for 24 h in normal growth medium (DMEM +10% FCS; Sigma). For *in vitro* stimulation experiments, cells were serum-starved overnight and pre-treated for 30 min with DMSO (1:1,000) or inhibitors against EGFR (5 μM afatinib, Santa Cruz Biotechnology), PDK-1 (10 μM GSK2334470, Selleckchem), mTor (10 nM rapamycin, Sigma), or ERK1/2 (10 μM U0126, Cell Signaling Technology). Afterward, cells were stimulated for 0, 5, or 30 min with 50 ng/ml recombinant EGF (PeproTech). All cell lines were tested and showed no mycoplasma contamination.

### Gene silencing using shRNA

To knock down (KD) *FOS*, HEK 293T cells were transduced with lentiviral *FOS* shRNA expression vectors (Tet-pLKO-neo-FOS) with pCMV-Δ89 plasmid and pVSV-G envelop plasmid, and then lentiviral particles from culture supernatants were used to infect 143b cells, which were selected with Neomycin (500 μg/ml) for 1 week.

shRNA oligo primer sequences for FOS were as follows: Fw: CCG GTC TGC TTT GCA GAC CGA GAT TCT CGA GAA TCT CGG TCT GCA AAG CAG ATT TTT G and Rv: AAT TCA AAA ATC TGC TTT GCA GAC CGA GAT TCT CGA GAA TCT CGG TCT GCA AAG CAG A.

### Immunofluorescence staining

For *in vitro* immunofluorescence (IF) staining, cells were seeded on chamber slides (2-well or 8-well) and cultured/treated as indicated in the figure legends. After removing the medium, cells were washed with PBS and fixed with 4% PFA for 20 min at room temperature. The fixed cells were washed and permeabilized with 0.1% Triton X-100 in PBS for 10 min at room temperature. Afterward, cells were washed and blocked with 2% BSA and 5% normal goat serum in PBS for 1 h at room temperature. Antibodies (see Appendix Table S1) were incubated in blocking buffer overnight at 4°C. After washing with PBS, 100 μl of secondary antibody (donkey anti-rabbit Alexa Fluor 555) diluted 1:500 in blocking buffer was added and incubated for an hour at room temperature. After washing with PBS, the slide was mounted with aqueous mounting medium and observed using a fluorescent microscope.

### MTT assay

Cells were seeded at a density of 500 cells/well in 96-well plates and cultured for 48, 96, and 144 h with/without doxycycline (500 ng/ml), respectively. Subsequently, 25 μl of MTT reagent (5 mg/ml) was added to each well and incubated for 2 h at 37°C. After removing media, wells were washed with PBS and then the remaining crystals were dissolved in dimethyl sulfoxide. The absorbance was evaluated at 570 nm (background is 630 nm). The assays were performed in triplicate.

### Cell viability assay

For cell viability assay, cells ($1.5 \times 10^3$/cm$^2$ for LM7; $1.5 \times 10^2$/cm$^2$ for 143b) were seeded in white, opaque-walled 96-well plates (Thermo Fisher). After 24 h, cells were treated with DMSO (1:2,000) or erlotinib (10 μM) in normal growth medium containing 10% FCS or stimulated with EGF (50 ng/ml) in growth medium containing 0.5% FCS. After 24 h of treatment/stimulation, cell viability was measured using the CellTiter-Glo® Luminescent Cell Viability Assay Kit (Promega) according to manufacturer's instructions.

### *In vivo* imaging

For X-ray analysis, mice were anesthetized and imaging was performed using a Mobilett XP mobile X-ray machine (Siemens). Tumor number was quantified from X-ray images using Photoshop CS6 (Adobe), and tumor size was measured with ImageJ 1.49v (Wayne Rasband, NIH, USA).

Imaging experiments were conducted with a small animal computed tomography (CT) and positron emission tomography (PET) scanner (Siemens Inveon™ multimodal μSPECT/PET/CT; Siemens Preclinical Solutions, Knoxville, USA). Mice were anesthetized with 1.5–2% isoflurane mixed with oxygen (1.5 l/min) to avoid movement during the imaging procedure. Subsequently, for the PET image acquisition mice were administered fluorine-18 sodium fluoride (Na[18F]F) (21.09 ± 3.86 MBq; 70 ± 19 μl) via retro-orbital injection. After an accumulation time of 60 min, a 10-min static PET acquisition and a subsequent high-resolution whole-body CT scan were performed. Vital parameters, such as respiration and body temperature, were recorded continuously using a dedicated physiological monitoring system (bioVet; m2imaging, Cleveland, OH, USA).

CT raw data were reconstructed with a Feldkamp algorithm using a Ramp filter followed by default beam-hardening correction. PET data reconstruction was performed with an OSEM 3D/OP-MAP algorithm with default parameters and attenuation corrected on the basis of CT measurements. All data were normalized and corrected for random, dead time, radioactive decay, and weight of the animal. A calibration factor was applied to the data in order to convert the units of the small animal PET images into absolute radioactivity concentrations, expressed as standardized uptake values (SUV; g/ml). PET and CT images were co-registered through an automatically generated spatial transformation matrix. Biomedical image analysis was accomplished using the Inveon Research Workplace Software (IRW; Siemens Medical Solutions, Knoxville, TN, USA). For image quantification, cylindrical volumes of interest were drawn. The respective Na[18F]F uptake in the delineated areas was expressed as SUV normalized to body weight.

PET experiments to study tissue distribution of [11C]erlotinib were conducted at the Austrian Institute of Technology on a microPET Focus 220 system (Siemens Medical Solutions) as described in detail elsewhere (Traxl *et al*, 2017). In brief, 6-month-old female H2-*c-fos*LTR/*Egfr*^wt mice (*n* = 3) underwent under isoflurane anesthesia 90-min dynamic PET scans after i.v. injection of a mixture of [11C]erlotinib (47 ± 12 MBq) and unlabeled erlotinib hydrochloride (Apollo Scientific, Bredbury, UK) at a dose of 2 mg/kg. At the end of the [11C]erlotinib PET scan, animals received

an i.v. injection of Na[$^{18}$F]F (injected amount: 5 ± 1 MBq) followed by a 20-min static PET scan. PET data were reconstructed as described (Traxl *et al*, 2017). Tumor volumes of interest were outlined on the dynamic [$^{11}$C]erlotinib PET images guided by the static Na [$^{18}$F]F PET images. Radioactivity concentration in tumors was expressed in SUV units and multiplied by the administered erlotinib dose of 2 mg/kg to obtain mass concentrations of erlotinib (μg/g).

### Intratibial tumor cell injection and *in vivo* erlotinib treatment

For intratibial injection of tumor cells, 8- to 12-week-old female nude mice were anesthetized and 143b or LM7 tumor cells ($10^6$ in 25 μl 1× PBS) were injected through the knee to the bone marrow cavity of the left proximal tibia. Seven days (143b) or 6 weeks (LM7) after injection, mice were randomized and treated for 14 days (143b) or 21 days (LM7) with erlotinib (50 mg/kg bw, i.p.) or the vehicle (0.5% Methylcellulose). Tumor volume was calculated by caliper measurements of the injected tibia bone using the formula Vol. = length × width$^2$/2 minus the volume of the healthy, non-injected leg as described elsewhere (Gvozdenovic *et al*, 2016). For long-term treatment, 2-month-old male H2-*c-fos*LTR mice were divided into two groups and treated with either erlotinib (25 mg/kg bw, 5×/week, i.p.) or the vehicle (0.5% Methylcellulose, 5×/week, i.p.; Politi *et al*, 2006). After 4 weeks of treatment, mice were left untreated for 4 weeks until the treatment regime started again. Tumor number and size as well as serum ALP levels were quantified monthly.

### Human mRNA expression data

For human mRNA expression data, the publicly available RNAseq datasets E-GEOD-39058 (Kelly *et al*, 2013) were downloaded from Gene Expression Omnibus and analyzed with the GEOquery package of R. The GSE39058 dataset also included recurrence-free survival information. Expression data were stratified into EGFR/FOS low and high expressers using the surv_cutpoint function of survminer using recurrence-free survival as target variable. The gene expression profile EGFR/FOS double low vs. double high was analyzed using the LIMMA package of R.

### Human OS tissue microarray

The human OS TMA was purchased from Novus Biologicals (Catalog Number: NBP2-30289). The TMA contained 4-μm-thick sections of 59 bone cancer biopsies and one control biopsy from a non-cancer patient. Seven patients had to be excluded from the analysis due to: unknown cause of death ($n = 2$), died of heart disease ($n = 2$), survival information not available ($n = 2$), and tissue lost during IHC staining procedure ($n = 1$). The remaining 52 biopsies, which were analyzed in the study, were derived from 39 male and 13 female patients with an average age of 20.2 ± 11.2 years.

### Immunohistochemistry

For immunohistological stainings, samples were fixed in 4% PBS-buffered formalin, decalcified in 0.5 M EDTA (pH 8.0) and embedded in paraffin. Immunohistochemistry was performed on 3-μm-thick sections. Primary antibodies (for a full list, see Appendix Table S1)

were incubated overnight at 4°C followed by HRP-based immunoreactivity detection (Cell Signaling Technology). Non-specific binding was blocked with TBS-T containing 2% BSA and 5% normal goat serum. Quantification of IHC stainings was performed using the ImageJ plugin ImmunoRatio (version: 1.0c, Jorma Isola and Vilppu Tuominen, University of Tampere). For analysis of EGFR/c-Fos expression in human OS, two consecutive sections from a commercially available TMA (Novus, NBP2-30289) were stained with antibodies directed against EGFR (Cell Signaling Technology, #4267) and c-Fos (Santa Cruz Biotechnology, SC-52). For quantification, cores were graded according to positive cells by two independent investigators whereby more than 60% positive cells were considered as positive/high. The results were afterward double-checked using an automatized histology quantification program (Definiens Tissue Studio$^®$ 4.0).

### RNA isolation, real-time qRT–PCR analysis

RNA isolation and qRT–PCR analysis were performed as described previously (Linder *et al*, 2018). For a full list of primers used, see Appendix Table S2.

### Western blot analysis

Western blot analysis was performed as previously described (Sibilia *et al*, 2000). For a full list of the antibodies used, please see Appendix Table S1.

### ELISA

Mouse Amphiregulin ELISA (Thermo Fisher) was performed according to manufacturer's instructions. Protein concentration was analyzed using Bradford protein assay (Bio-Rad).

### Analysis of alkaline phosphatase

Alkaline phosphatase serum levels were measured using diagnostic ALP test stripes (Reflotron, Roche) and read on an automatic Reflotron analyzer (Roche).

### Statistical methods

Sample size of *in vivo* experiments was chosen based on previous experience with similar studies in the laboratory. For analyses of IHC and qRT–PCR data, univariable comparisons of expression values between groups were analyzed by unpaired two-tailed Student's *t*-test with *f*-test to ensure comparable variances between the groups. If the variances were significantly different, as measured by *f*-test, unpaired two-tailed Student's *t*-test with Welch's correction was performed. For analysis of more than two groups/time-points (e.g., erlotinib long-term treatment experiment), two-way ANOVA with Bonferroni post-test was applied. For comparison of survival curves, log-rank (Mantel–Cox) tests were performed. Data are shown as mean ± SEM. A *P*-value below 0.05 was considered statistically significant. For analyses and calculation, GraphPad Prism 6 software was used.

**Expanded View** for this article is available online.

## The paper explained

### Problem

Osteosarcoma (OS) is a rare tumor affecting the bone and occurs mainly in children and young adults. Besides surgery and chemotherapy, there are no other therapeutic options and there has been no survival improvement for OS patients in the last 40 years. The epidermal growth factor receptor (EGFR) is highly expressed in OS patients. However, because of the low incidence of OSs, clinical studies with EGFR inhibitors could so far not demonstrate a clear therapeutic benefit.

### Results

Here, we report that EGFR promotes OS development via a signaling pathway involving RSK2/CREB leading to upregulation of the AP-1 transcription factor c-Fos. Genetic deletion or pharmacological inhibition of EGFR resulted in reduced OS formation and progression in mouse models of OS by inhibiting the proliferation and survival of cancer-initiating osteoblastic cells. Furthermore, we found that patients suffering from OSs, which are positive for both EGFR and c-Fos, show significantly reduced overall survival. Preclinical studies using orthotopic human OS Xenografts revealed that only human OSs co-expressing EGFR and FOS responded to anti-EGFR therapy, whereas tumors that were EGFR-positive but EGFR-negative for FOS did not respond.

### Impact

Our results demonstrate that c-Fos and EGFR represent novel biomarkers predicting response to anti-EGFR treatment in OS patients. Thus, our findings will allow better patient stratification in the future and our study recommends targeted anti-EGFR therapies for OS patients positive for Fos and EGFR.

## Acknowledgements

We thank Martina Hammer for maintaining mouse colonies, Malgorzata Tryniecki for genotyping, Bilge Vasfiye Göcen for technical assistance, and Andrea Nolz for help with X-ray imaging. We are grateful to Marlon R. Schneider for providing the *ColAREG* transgenic mice and to Eugenie S. Kleinerman for providing the LM7 OS cell line. This work was supported by the Austrian Science Fund (FWF) (DK W1212). M.S. is supported by an ERC-Advanced grant (ERC-2015-AdG TNT-Tumors 694883). The [$^{11}$C]erlotinib PET imaging part of this work was supported by the FWF project "Transmembrane Transporters in Health and Disease" (SFB F35).

## Author contributions

ML performed and analyzed most experiments. EG, SS, LB, KM, PS, OL, TW, PN, MM, and MD performed experiments. TM performed bioinformatic analysis of the RNAseq dataset. LB and EFW provided essential input into the experimental design and helped with data analysis and interpretation. MS conceived and supervised the project and provided the funding. ML and MS wrote the manuscript with input from the other authors. All authors analyzed the results and approved the final version of the manuscript.

## Conflict of interest

The authors declare that they have no conflict of interest.

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
