## [Review Process File · EMBO Molecular Medicine]

EGFR is required for FOS-dependent bone tumor development via RSK2/CREB signaling

Markus Linder, Elisabeth Glitzner, Sriram Srivatsa, Latifa Bakiri, Kazuhiko Matsuoka, Parastoo Shahrouzi, Monika Dumanic, Philipp Novoszel, Thomas Mohr, Oliver Langer, Thomas Wanek, Markus Mitterhauser, Erwin F. Wagner and Maria Sibilica

Review timeline:

Submission date:	08 June 2018
Editorial Decision:	06 July 2018
Revision received:	28 August 2018
Editorial Decision:	20 September 2018
Revision received:	24 September 2018
Accepted:	25 September 2018

Editor: Lise Roth

Transaction Report:

1st Editorial Decision

06 July 2018

Thank you for the submission of your manuscript to EMBO Molecular Medicine. We have now heard back from the three referees whom we asked to evaluate your manuscript.

As you will see from the reports below, the referees are overall positive and support publication of the article in EMBO Molecular Medicine pending appropriate revisions. However, while they acknowledge the technical quality and interest of the study, they also agree that the medical impact could be increased pending additional analyses and experiments (EGFR/c-Fos expression in patients' samples, late death/survival analyses). Other important points include clarification between endogenous and transgenic c-Fos, characterization of the 143b cell line, addition of appropriate control and thorough discussion. Addressing the reviewers concerns in full will be necessary for further considering the manuscript in our journal. EMBO Molecular Medicine encourages a single round of revision only and therefore, acceptance or rejection of the manuscript will depend on the completeness of your responses included in the next, final version of the manuscript.

Please also contact us as soon as possible if similar work is published elsewhere. If other work is published, we may not be able to extend the revision period beyond three months.

I look forward to receiving your revised manuscript.

***** Reviewer's comments *****

Referee #1 (Comments on Novelty/Model System for Author):

The data presented and the models used are of the highest technical quality. This report comes from a highly credentialed laboratory and I have no issues with technical quality.

I have suggested that novelty and medical impact are medium. This is mainly because there is some other data from other groups and clinical trials published. Thus, the content of this manuscript is novel but is probably incremental rather than paradigm shifting.

The medical impact is medium. It may have the potential to be translated but there is no clear pathway to translation at present. I think if the authors address my last comment about EGFR/c-fos expression in their patient samples and Kaplan Meir curves this could pave the way to a trial in chemorefractory patients. This is not an easy trial to get up given 60-70% of patients have good outcomes and so your dealing with an ever reducing population of patients to trial. Unless of course all chemorefractory patients are EGFR/c-fos positive in which case this becomes a very interesting proposition.

Referee #1 (Remarks for Author):

Linder et al provide data supporting a role for EGFR as a driver of c-fos-dependent osteosarcoma proliferation. They provide in vitro, in vivo and patient data sets to support this argument. They go on to suggest that EGFR/c-fos expression levels could be used to identify patients for treatment with small molecular weight inhibitors of EGFR. Overall, they make a good case for this and the data are supportive of this conclusion.

I have a few comments (major and minor) that the authors may wish to consider in a revised manuscript.

In Fig 1 and throughout the manuscript the authors provide data that EGFR/c-fos dependent tumour promotion is due to enhanced proliferation. To this end, inhibition of EGFR signalling reduced tumour growth. It would have been informative if the authors had also examined markers of apoptosis. This would allow them to ascribe their effects entirely to proliferation inhibition or possibly to reduced proliferation and enhanced apoptosis. At present the contribution of cell death to reduced tumour growth is presumed, yet not proven, to be negligible. This could be an important issue as the use of a purely cytostatic therapy in paediatric cancers may not be as readily embraced as a drug which holds the potential for cytotoxicity (especially if part of combination therapy).

At the bottom of 7 the authors state "These data indicate that EGFR signaling in osteoblasts is not only essential for proliferation, but also controls endogenous c-fos expression and RSK2/pCREB/c-Fos protein stabilization in c-Fos-dependent OSs." What is the evidence for protein stabilization. Can they rule out other events such as changes to phosphatase activity.

On page 8 the authors make the comment ".....leading to reduced proliferation as shown by reduced cyclinD1 protein levels". The use of PCNA, Ki67 or BrdU would provide stronger support for this statement rather than reliance on cyclinD1 levels.

The authors use a transgenic mouse model which expresses tg c-fos in the context of endogenous c-fos expression. Whilst this tumour model requires tg c-fos, the authors make it clear that the EGFR-dependent effects on tumour proliferation are due to the endogenous c-fos. This is quite a complicated concept and perhaps requires a little more attention to clarify in the text. For example, if c-fos expression is dependent upon EGFR stimulation (Fig 4E) would this suggest we should see c-fos activation in all patient tumours and experimental samples in which EGFR is expressed (S6A and B data). The same applies to the cell lines LM7 and 143b which both express EGFR yet only the 143b expresses c-fos. This may require a little more clarification.

Based on the data in Fig 6B and S6A & B I assume that 13/52 (25%) of patient tumours would have EGFR/c-fos expression. It may be helpful to interrogate the EGFR/c-fos expressing samples to find out if the patients had chemo-refractory disease or not. OS biopsy specimens are taken after the

initial cycles of chemo and a poor chemo response should be noted in the Pathologist report (tumour kill < 90%) and is prognostic of poor outcome. It may be likely that the c-fos/EGFR patients in Fig 6B correspond with this. This could be incredibly helpful as it is this patient cohort who can benefit most from new therapies.

Referee #2 (Comments on Novelty/Model System for Author):

The medical impact could reach high if they perform the analyses that I suggested.

Referee #2 (Remarks for Author):

The manuscript describes the results of a comprehensive analysis of the role of EGFR in Fos-dependent bone tumor formation. Their results should interest a wide audience and potentially have clinical significance.

The transgenic strain which is the basis of the work presented in the manuscript under review, Linder et al, is the H2-c-fosLTR c-fos in which the ubiquitously expressed H-2K b class I MHC promoter drives c-Fos expression, yielding bone tumors with 100% penetrance. My search of the literature suggested that this strain was first reported on and well characterized in Agamemnon et al JCB 1993. Thus, on pg 3 of the manuscript I am not sure why this reference is not mentioned, in addition to Ruther et al 1989. In the JCB paper they show that expression of the introduced c-fos gene was high in the bone-forming osteoblastic cells and in the resulting tumors. Since the penetrance is essentially 100% and the tumors develop rapidly, it appears that c-Fos is the driving oncogene.

With this as background, the current paper examined the potential role of EGFR in OS. EGFR was chosen since on the one hand, the same group recently published on the essential role of EGFR in osteoblasts for bone development. Moreover, it has been reported by various groups that up to 80% of tumors express the receptor. A quick check of the older literature suggests that in OS EGFR amplification occurs rarely, if at all, so this is unlikely to be the reason for elevated levels of the receptor. There are also not many reports of EGFR activating mutations in OS. There have been a few clinical trials looking at the effect of blocking EGFR with kinase inhibitors or with a blocking antibody. These have not yielded conclusive results. It is well known that overexpression of EGFR in the absence of a genetic alteration cannot be used as a predictive marker for EGFR-directed therapy response in cancer. This is one reason that the trials may have given negative or inconclusive results. The new results in this manuscript do suggest that another look at targeting EGFR in OS would be warranted. I have some suggestions as to how the manuscript could be improved.

General comments and questions

1. They show 1st that crossing the H2-c-fosLTR c-fos Tg with either Egfr waved2 mice or treatment with the kinase inhibitor erlotinib has a strong negative impact on the number and growth of bone tumors in the mice. Fig S1D shows that EGFR is phosphorylated in tumors from vehicle treated mice and that erlotinib-treated mice show a strong decrease in P-EGFR. Thus, in the H2-c-fosLTR c-fos Tg model, the results suggest that high levels of constitutive c-fos causes "changes" that result in a constitutive loop between EGFR and c-Fos, since P-EGFR is present. Does the receptor become active in these tumors due to autocrine production of an EGF ligand? What ligands are produced in the bone tumors? The data showing that overexpression of the AREG Tg together with the H2-c-fosLTR c-fos Tg accelerates OS development (Fig 5) support this idea. It would be good know what ligands are expressed in the bone tumors from the single transgenics.

2. In the next set of experiments they showed the EGFR expression in the osteoblastic lineage and not in the myeloid lineage is required for robust OS development. They then turned to more mechanistic studies showing that there is less proliferation and decreased cyclinD1 levels in the tumors from the Egfr deltaOb strain. Interestingly, these tumors also show lower levels of endogenous c-Fos mRNA, but not the Tg c-Fos mRNA (Fig 3B), showing that EGFR is needed to maintain endogenous c-Fos RNA levels. The c-Fos protein levels were also significantly lower (Fig 3C). Various signaling proteins were also down-regulated including pCREB and pRSK2, and the

amount of phosphorylation on c-Fos Ser362 was decreased. This site is important for c-Fos stabilization.

Question - Does the finding that there is less total c-Fos protein mean that both the endogenous and the Tg protein are affected? It is not clear to me from the discussion whether or not all the c-Fos in the OS cells, the tg and the endogenous c-Fos, would be affected by loss of P-Ser362? Please clarify.

3. The data presentation in Fig 4 and S4 is not optimal. They start by discussing the pRSK2/pCREB/c-Fos axis and in Fig 4C show data on the Phospho- status of these proteins after erlotinib treatment. They then use a panel of afatinib and other inhibitors affecting different pathways and show the data in Fig 4E and Fig S4B. Why is there is no data on the pRSK status presented? Why did they switch EGFR inhibitors? It would be better if all these data were in the main figures of the paper and the results with pRSK were shown.

4. The authors aim to make the results with the GEM models applicable for the clinic, which is very positive since as they rightly mention there has been very little progress in therapy for bone tumors. For this they used a panel of 52 human OS tumors and stained for EGFR and c-Fos levels (Fig 6A). EGFR or c-Fos levels on their own did not significantly correlate with overall survival of the patients (Fig S6A and B). Since the overall survival rate in this cancer is usually expressed in terms of 5-years, I suggest that they show the data plotted in a 60 month curve. If they do this, is there a significant difference between OS survival and high levels of EGFR or high levels of c-Fos alone? Irrespective of this outcome, if you look at the time after 5 years it is interesting that many of the low EGFR or low c-Fos patients continue to survive, while the high group continues to die. In Fig 6, it appears that when both proteins are low, there is a plateau starting at about 3 years showing that about 60% of the patients survive >150 months. The double positive group continues to lose patients over 100 months, which reflects the single curves in Fig S6A-B.

Do the authors have any data on p-EGFR levels in the 52 OS tumors? That would allow a further stratification.

My suggestion - if the data are looked at in terms of early or late death/survival, are the conclusions the same? Are the patients with high levels of both EGFR and c-Fos showing a significant decrease in overall survival at 5-years? What about the 5-year OS correlation with high EGFR or high c-Fos alone?

I ask this in light of the next question/comment.

5. Putting the results of the H2-c-fosLTR c-fos model into the human context, is it possible to speculate that another "driver" turns on a pathway that maintains constitutive c-Fos expression? Maybe c-Fos levels are what should be looked at in the clinic.

In the Tg c-Fos model this additional "driver" would be active EGFR, but what about in OS patients? Is it possible that there are high levels of expression of other RTKs that lie upstream of c-Fos? Did the authors mine the available cancer data sets, TCGA etc, for tumors with elevated c-Fos levels alone then combine this with other potential drivers including cancer-associated RTKs or other signaling molecules that can impact on c-Fos expression levels? I think that the paper would be more clinically relevant if some of these points could be addressed and discussed in the paper.

6. Their work with the human cell lines (Fig 7) is interesting since it shows that the one line (143b) with high levels of both EGFR & c-Fos is more sensitive to the kinase inhibitor. There is a strong decrease in p-c-Fos as well in response to erlotinib.

Do the authors have any data or ideas as to on what is driving the elevation of EGFR? Does this cell line have EGFR amplification? If c-Fos is deleted, do the cells die?

Have the authors waited longer with the models before starting treatment with the inhibitor? In other words, would the tumors regress if treatment were begun on day 8 instead of day 0 for the 143b model?

7. Many of the legends to the Supplementary Figures are missing. This needs to be carefully checked in the text as well.

Minor points

P19 typo - "..were injected through.." not though

P 31 typo - Fig 6B legend "survival of patients" not patient.

P31- In the legend to Fig 7 panel B-C the figure is labeled cell viability (%). This should be labeled as an "increase in the number of viable cells", since a cell culture that is 100% viable cannot be made more viable- 150%.

P23 - In the author contribution section, EFW is not specifically mentioned.

Referee #3 (Comments on Novelty/Model System for Author):

Authors use an established genetically-engineered mouse model for induction of osteosarcoma with features of the human disease.

Referee #3 (Remarks for Author):

Osteosarcoma (OS) is an uncommon cancer affecting children and young adults. Metastatic osteosarcoma has a poor prognosis. Loss of p53 and Rb are common in OS, but the genetic driver landscape is otherwise complex. The authors have identified a role for Egfr in mouse bone development, and other reports describe high EGFR expression in a majority of human OS. A genetically engineered mouse model (H2-c-fosLTR) that induces high levels of c-fos induces OS. The authors report that endogenous egfr activity is important in activation of c-fos in this model, working through MAPK to CREB and Fos to induce OS. In vitro and in vivo inhibition or genetic loss of Egfr reduce OS formation and growth, concomitant with reduction in RSK2, Fos, and CREB signaling, reduced Cyclin D1, and reduced entry into cell cycle. This pathway is inhibited by MAPK inhibitor, but not PI3K pathway inhibitors. Over 200 weeks, high OS expression of FOS plus EGFR foreshadow poorer prognosis. EGFR-dependent cell line 143b but not EGF-insensitive line LM7, is growth-inhibited by erlotinib in xenografts. The authors conclude that EGFR inhibitors may be useful for control of a substantial subset of OS with high EGFR and FOS.

This is a clearly-written manuscript that attacks an important and challenging clinical problem. The data mainly well-controlled and convincing. The biological connection of EGFR to OS growth in the fosLTR model is interesting but not entirely surprising (although many other receptors could potentially have turned out to be involved.) An important finding is the connection of EGFR/FOS with prognosis in human OS in the context of the mouse model and xenograft results. The major biological issues are the relationship of the Fos model to human OS. More information on the OS biology in the context of these findings would enhance an already solid manuscript.

1. The authors recently reported the importance of EGFR in bone development. This means that the cellular landscape of bone and OS tissue in the egfr^{-/-} model may be very different from WT. In Fig. 3 and other related figures, does the loss of egfr and fos immunostaining reflect osteoblast cellularity or differentiation? Would this affect the frequency of cell-of-origin for FOS-driven OS, or, as the authors describe, is this mainly a signaling issue for activation of FOS.

2. Similarly, do double EGFR⁺ FOS⁺ human OS have different features (e.g. transcriptional profiles, putative genetic or epigenetic drivers) than the double negatives? This information may be available in public datasets.

3. There is a great range of expression of EGFR in the six OS lines tested. Is high EGFR in 143b cell and in human OS associated with gene amplification? What are candidate ligands for EGFR in human OS?

4. The partial inhibition of 143b xenografts by EGFR TKI is clear, but this level of response, with continued tumor growth, is not very strong for a xenograft model, and may suggest that EGFR inhibitors would not work well as single agents.

Minor issues.

1. The blots and IHC are mostly excellent, but specificity is an issue of concern for IHC. How were human EGFR and FOS and other immunostaining validated?

2. No positive control to show that rapamycin and GSK agent work.

3. "EGFR addiction can be harnessed therapeutically to longlasting stable disease" may be over-optimistic. In EGFR lung adenocarcinoma, EGFR inhibitors have yet to control disease beyond 24 or so months on average time to progression, and worse for other diseases.

1st Revision - authors' response

28 August 2018

Referee #1 (Comments on Novelty/Model System for Author):

The data presented and the models used are of the highest technical quality. This report comes from a highly credentialed laboratory and I have no issues with technical quality.

I have suggested that novelty and medical impact are medium. this is mainly because there is some other data from other groups and clinical trials published. Thus, the content of this manuscript is novel but is probably incremental rather than paradigm shifting.

The medical impact is medium. It may have the potential to be translated but there is no clear pathway to translation at present. I think if the authors address my last comment about EGFR/c-fos expression in their patient samples and Kaplan Meir curves this could pave the way to a trial in chemorefractory patients. This not an easy trial to get up given 60-70% of patient have good outcomes and so your dealing with an ever reducing population of patients to trial. Unless of course all chemorefractory patients are EGFR/c-fos positive in which case this becomes a very interesting proposition.

Many thanks for appreciating the high technical quality of our manuscript. We have addressed all her/his concerns as specified below.

Referee #1 (Remarks for Author):

Linder et al provide data supporting a role for EGFR as a driver of c-fos-dependent osteosarcoma proliferation. They provide in vitro, in vivo and patient data sets to support this argument. They go on to suggest that EGFR/c-fos expression levels could be used to identify patients for treatment with small molecular weight inhibitors of EGFR. Overall, they make a good case for this and the data are supportive of this conclusion.

I have a few comments (major and minor) that the authors may wish to consider in a revised manuscript.

In Fig 1 and throughout the manuscript the authors provide data that EGFR/c-fos dependent tumour promotion is due to enhanced proliferation. To this end, inhibition of EGFR signalling reduced tumour growth. It would have been informative if the authors had also examined markers of apoptosis. This would allow them to ascribe their effects entirely to proliferation inhibition or possibly to reduced proliferation and enhanced apoptosis. At present the contribution of cell death to reduced tumour growth is presumed, yet not proven, to be negligible. This could be an important issue as the use of a purely cytostatic therapy in paediatric cancers may not be as readily embraced as a drug which holds the potential for cytotoxicity (especially if part of combination therapy).

We apologize for not having mentioned apoptosis in our previous manuscript. In the revised paper, we now provide evidence that EGFR is essential for inducing survival not only in the H2-c-fosLTR transgenic mouse model (see Fig. 3A, Fig. EV3A, Fig. 5G) but also in orthotopic tumors (see Fig. 7J). Therefore, therapeutic approaches targeting EGFR have a potential for cytotoxicity.

At the bottom of 7 the authors state "These data indicate that EGFR signaling in osteoblasts is not only essential for proliferation, but also controls endogenous c-fos expression and RSK2/pCREB/c-Fos protein stabilization in c-Fos-dependent OSs." What is the evidence for protein stabilization. Can they rule out other events such as changes to phosphatase activity.

This is a very valid point. We cannot rule out that other events such as changes in phosphatase activity affect c-Fos protein stabilization., However, there is evidence that RSK2 is essential to

maintain c-Fos protein (but not mRNA) expression and stabilization via RSK2-dependent phosphorylation of c-Fos at Serine 362 in H2-c-fosLTR mice (David et al, 2005). We also found reduced pRSK/c-Fos protein levels along with reduced phosphorylation of c-Fos on Serine 362 when EGFR was deleted in the H2-c-fosLTR mouse model (Fig. EV2G) and in xenografts after Erlotinib treatment (see Fig. 7L) indicating that EGFR acts as the upstream receptor affecting c-Fos stabilization at least via pRSK2.

On page 8 the authors make the comment ".....leading to reduced proliferation as shown by reduced cyclinD1 protein levels". The use of PCNA, Ki67 or BrdU would provide stronger support for this statement rather than reliance on cyclinD1 levels.

We have performed immunofluorescence analysis for Ki67 and are showing the quantification in Extended View Figures EV3B & D. We found that EGFR is essential for proliferation as shown by reduced Ki67 positive cells in both OS cells that lacked EGFR and in cells that were treated with the EGFR inhibitor Erlotinib (Figs. EV3B, D).

The authors use a transgenic mouse model, which expresses tg c-fos in the context of endogenous c-fos expression. Whilst this tumour model requires tg c-fos, the authors make it clear that the EGFR-dependent effects on tumour proliferation are due to the endogenous c-fos. This is quite a complicated concept and perhaps requires a little more attention to clarify in the text. For example, if c-fos expression is dependent upon EGFR stimulation (Fig 4E) would this suggest we should see c-fos activation in all patient tumours and experimental samples in which EGFR is expressed (S6A and B data). The same applies to the cell lines LM7 and 143b which both express EGFR yet only the 143b expresses c-fos. This may require a little more clarification

The reviewer is absolutely right. To clarify this point, we included additional experiments with primary OS cells isolated from another transgenic OS mouse model, namely the p53^{f/f} Rb1^{f/f} Osx-Cre strain. In this model, it has been shown that endogenous c-Fos is up-regulated in primary OS cells as compared to isolated osteoblasts (Walkley et al, 2008). We found that also in this model EGFR is essential for c-Fos mRNA and protein expression via RSK2/CREB signaling indicating that the mechanism proposed here applies also for OSs driven by the loss of tumor suppressors (see Figs. 4G-I). When we performed EGF stimulation experiments in LM7 cells, we could not see any upregulation of c-Fos indicating that EGFR-dependent c-Fos activation requires at least basal levels of endogenous c-Fos expression (Fig. EV5B). This mechanism does not seem to operate in completely FOS negative tumor cells like LM7. This would also explain why some patients show expression of EGFR but not c-Fos (Figs. 6B and EV4A, B).

Based on the data in Fig 6B and S6A & B I assume that 13/52 (25%) of patient tumours would have EGFR/c-fos expression. It may be helpful to interrogate the EGFR/c-fos expressing samples to find out if the patients had chemo-refractory disease or not. OS biopsy specimens are taken after the initial cycles of chemo and a poor chemo response should be noted in the Pathologist report (tumour kill < 90%) and is prognostic of poor outcome. It may be likely that the c-fos/EGFR patients in Fig 6B correspond with this. This could be incredibly helpful as it is this patient cohort who can benefit most from new therapies.

It would indeed be interesting to evaluate whether EGFR/c-Fos expression status correlates with chemo-refractory disease or not. However, as we used commercially available TMAs to analyze c-Fos / EGFR expression levels, we only have the survival data of the patients with no additional clinical information. We contacted the company to ask for more information. Unfortunately, they do not have information on the treatment of patients. Therefore, regrettably we cannot address this issue.

Referee #2 (Comments on Novelty/Model System for Author):

The medical impact could reach high if they perform the analyses that I suggested.

Referee #2 (Remarks for Author):

The manuscript describes the results of a comprehensive analysis of the role of EGFR in Fos-dependent bone tumor formation. Their results should interest a wide audience and potentially have clinical significance.

We would like to thank the reviewer for her/his comments and for appreciating the potential interest of our results to a wide audience and the clinical relevance of our findings.

The transgenic strain which is the basis of the work presented in the manuscript under review, Linder et al, is the H2-c-fosLTR c-fos in which the ubiquitously expressed H-2K b class I MHC promoter drives c-Fos expression, yielding bone tumors with 100% penetrance. My search of the literature suggested that this strain was first reported on and well characterized in Agamemnon et al JCB 1993. Thus, on pg 3 of the manuscript I am not sure why this reference is not mentioned, in addition to Ruther et al 1989. In the JCB paper they show that expression of the introduced c-fos gene was high in the bone-forming osteoblastic cells and in the resulting tumors. Since the penetrance is essentially 100% and the tumors develop rapidly, it appears that c-Fos is the driving oncogene.

We apologize for not having cited the paper by Agamemnon Grigoriadis et al, 1989 on page 3, line 56-58. We included it now in the revised manuscript.

With this as background, the current paper examined the potential role of EGFR in OS. EGFR was chosen since on the one hand, the same group recently published on the essential role of EGFR in osteoblasts for bone development. Moreover, it has been reported by various groups that up to 80% of tumors express the receptor. A quick check of the older literature suggests that in OS EGFR amplification occurs rarely, if at all, so this is unlikely to be the reason for elevated levels of the receptor. There are also not many reports of EGFR activating mutations in OS. There have been a few clinical trials looking at the effect of blocking EGFR with kinase inhibitors or with a blocking antibody. These have not yielded conclusive results. It is well known that overexpression of EGFR in the absence of a genetic alteration cannot be used as a predictive marker for EGFR-directed therapy response in cancer. This is one reason that the trials may have given negative or inconclusive results.

The new results in this manuscript do suggest that another look at targeting EGFR in OS would be warranted. I have some suggestions as to how the manuscript could be improved.

General comments and questions

1. They show 1st that crossing the H2-c-fosLTR c-fos Tg with either Egfr waved2 mice or treatment with the kinase inhibitor erlotinib has a strong negative impact on the number and growth of bone tumors in the mice. Fig S1D shows that EGFR is phosphorylated in tumors from vehicle treated mice and that erlotinib-treated mice show a strong decrease in P-EGFR. Thus, in the H2-c-fosLTR c-fos Tg model, the results suggest that high levels of constitutive c-fos causes "changes" that result in a constitutive loop between EGFR and c-Fos, since P-EGFR is present. Does the receptor become active in these tumors due to autocrine production of an EGF ligand? What ligands are produced in the bone tumors? The data showing that overexpression of the AREG Tg together with the H2-c-fosLTR c-fos Tg accelerates OS development (Fig 5) support this idea. It would be good know what ligands are expressed in the bone tumors from the single transgenics.

We checked the mRNA expression levels of the EGFR ligands in H2-c-fosLTR transgenic mice and in an available RNAseq dataset of OS patients and found that Hb-EGF was the only ligand that was significantly upregulated in both mouse and men (Figs. EV1E, F). Furthermore, in mice we found that Hb-EGF and TGFA were the only ligands that were significantly upregulated in OS as compared to normal adjacent bones indicating that the tumors might indeed activate EGFR via autocrine production of these two ligands (see Figs. EV1G, H).

2. In the next set of experiments they showed the EGFR expression in the osteoblastic lineage and not in the myeloid lineage is required for robust OS development. They then turned to more

mechanistic studies showing that there is less proliferation and decreased cyclinD1 levels in the tumors from the Egfr deltaOb strain. Interestingly, these tumors also show lower levels of endogenous c-Fos mRNA, but not the Tg c-Fos mRNA (Fig 3B), showing that EGFR is needed to maintain endogenous c-Fos RNA levels. The c-Fos protein levels were also significantly lower (Fig 3C). Various signaling proteins were also down-regulated including pCREB and pRSK2, and the amount of phosphorylation on c-Fos Ser362 was decreased. This site is important for c-Fos stabilization.

Question - Does the finding that there is less total c-Fos protein mean that both the endogenous and the Tg protein are affected? It is not clear to me from the discussion whether or not all the c-Fos in the OS cells, the tg and the endogenous c-Fos, would be affected by loss of P-Ser362? Please clarify.

This is difficult to answer since the c-Fos protein expressed from the transgene is not tagged and cannot be distinguished from the endogenous c-Fos. We can only distinguish transgenic c-fos mRNA from total c-fos mRNA and we show that the levels of transgenic c-fos were not significantly changed between the genotypes, whereas total c-fos mRNA levels were reduced in the absence of EGFR, suggesting that EGFR is essential for endogenous c-fos, but not transgenic c-fos mRNA expression, which is not controlled from its natural promoter. (see Figs. 3B, 4A, 4D). Since the RNA levels reflect quite well the protein levels, we can indirectly assume that in the absence of EGFR the endogenous c-Fos protein is also reduced but cannot rule out the effect to the tg c-Fos protein. We also discussed this important point in the revised manuscript on page 17, line 367-370.

In the revised paper, we have performed several experiments with primary OS cells from the p53^{fl/fl} Rb1^{fl/fl} Osx-Cre mouse model and found that the same mechanism applies also for these cells (where there is no transgenic c-Fos). We see reduced pRSK2/CREB/c-Fos protein and reduced c-fos mRNA expression after Erlotinib treatment and increased RSK2/CREB/c-Fos signaling upon EGF stimulation – (see Figs 4G-I) – indicating that the mechanism described here does not depend on transgenic c-Fos but rather on endogenous c-Fos.

3. The data presentation in Fig 4 and S4 is not optimal. They start by discussing the pRSK2/pCREB/c-Fos axis and in Fig 4C show data on the Phospho- status of these proteins after erlotinib treatment. They then use a panel of afatinib and other inhibitors affecting different pathways and show the data in Fig 4E and Fig S4B. Why is there is no data on the pRSK status presented? Why did they switch EGFR inhibitors? It would be better if all these data were in the main figures of the paper and the results with pRSK were shown.

We included now pRSK2 in the Figure (Fig. 4E). We initially performed our in vitro experiments using the pan ErbB inhibitor Afatinib, then used the more EGFR-specific inhibitor Erlotinib. However, as EGF stimulation specifically activates EGFR, we used Afatinib as a control in the stimulation experiments. As suggested by the reviewer, we also moved the “control” WB, where all the other pathways are shown, to the main figure of the paper (Fig. 4F).

4. The authors aim to make the results with the GEM models applicable for the clinic, which is very positive since as they rightly mention there has been very little progress in therapy for bone tumors. For this they used a panel of 52 human OS tumors and stained for EGFR and c-Fos levels (Fig 6A). EGFR or c-Fos levels on their own did not significantly correlate with overall survival of the patients (Fig S6A and B). Since the overall survival rate in this cancer is usually expressed in terms of 5-years, I suggest that they show the data plotted in a 60 month curve. If they do this, is there a significant difference between OS survival and high levels of EGFR or high levels of c-Fos alone?

The figure provided in the paper shows the survival for up to 15 years. We performed an analysis for the 5-year survival as suggested by the reviewer and are showing this analysis in the figure below. Although there is a tendency that EGFR and Fos double positive patients show reduced overall survival after 5 years, this difference did not reach significance likely due to the low number of patients:

Irrespective of this outcome, if you look at the time after 5 years it is interesting that many of the low EGFR or low c-Fos patients continue to survive, while the high group continues to die. In Fig 6, it appears that when both proteins are low, there is a plateau starting at about 3 years showing that about 60% of the patients survive >150 months. The double positive group continues to lose patients over 100 months, which reflects the single curves in Fig S6A-B. Do the authors have any data on p-EGFR levels in the 52 OS tumors? That would allow a further stratification.

We analyzed pEGFR in the 52 OSs and found that half of the EGFR positive OS were also positive for pEGFR (Y1068). Although there was a tendency, survival analysis looking at Y1068 EGFR phosphorylation alone or in combination with c-Fos did not reach significance likely due to low patient samples. Of note, EGFR can be phosphorylated on five more tyrosine residues and as we don't know the activation status of these phosphorylation sites (no good antibodies are available to detect these sites), the analysis of only 1068 phosphorylation is to our opinion not very informative. We decided not to include these results in the final manuscript but are showing the analysis in the following figure for the reviewer:

My suggestion - if the data are looked at in terms of early or late death/survival, are the conclusions the same? Are the patients with high levels of both EGFR and c-Fos showing a significant decrease in overall survival at 5-years? What about the 5-year OS correlation with high EGFR or high c-Fos alone? (see comments above)

I ask this in light of the next question/comment.

5. Putting the results of the H2-c-fosLTR c-fos model into the human context, is it possible to speculate that another "driver" turns on a pathway that maintains constitutive c-Fos expression? Maybe c-Fos levels are what should be looked at in the clinic.

This is a very interesting point. Of course, another driver is possible since c-Fos is the target of many signaling pathways. However, in the analyzed TMA the levels of c-Fos protein expression alone did not significantly correlate with patient survival, both at late time points and when we looked at the 5-year survival only. On the other hand, high EGFR/c-Fos co-expression significantly correlated with reduced patient overall survival. This, together with the data from our mouse experiments, indicates that EGFR AND c-Fos co-expression status would be a better prognostic marker than c-Fos alone and more potentially, could predict treatment response to anti-EGFR therapy.

In the Tg c-Fos model this additional "driver" would be active EGFR, but what about in OS patients? Is it possible that there are high levels of expression of other RTKs that lie upstream of c-Fos? Did the authors mine the available cancer data sets, TCGA etc, for tumors with elevated c-Fos levels alone then combine this with other potential drivers including cancer-associated RTKs or other signaling molecules that can impact on c-Fos expression levels? I think that the paper would be more clinically relevant if some of these points could be addressed and discussed in the paper.

As suggested by the reviewer, we mined available osteosarcoma datasets and checked expression levels of other, cancer-associated RTKs and EGFR ligands in 42 human OS. We found that, besides EGFR, only PDGFRB significantly correlated with FOS expression (Fig. 6C). This raises the possibility that PDGFRB might also drives c-Fos expression during OS formation. We discussed this point also in page 18, line 395-401.

6. Their work with the human cell lines (Fig 7) is interesting since it shows that the one line (143b) with high levels of both EGFR & c-Fos is more sensitive to the kinase inhibitor. There is a strong decrease in p-c-Fos as well in response to erlotinib.

Do the authors have any data or ideas as to on what is driving the elevation of EGFR? Does this cell line have EGFR amplification?

We checked available datasets and found that 143B was included in the OncoMap 3.0 of the CCLLE and that EGFR was sequenced. However, no mutation of the EGFR was described for this cell line. In addition to that, Wen et al also performed sequencing of exons 18 to 21 of the EGFR gene in the 143b cell line and also found not functional mutation (Wen et al, 2007). We also provide this information now in the revised version of the manuscript (see page 20, line 434-435).

If c-Fos is deleted, do the cells die?

Thank you for addressing this important point! For the revised manuscript, we knocked down c-fos in 143b cells using a doxycycline-inducible shRNA approach (Figs. EV5C-H). 143b cells showed significantly reduced cumulative cell number, cell viability and cell proliferation when shRNA against c-fos was expressed. In addition, we also checked apoptosis and found a tendency of increased cell death in 143b OS cells lacking c-fos, which however did not reach significance, indicating that decreasing c-Fos expression mainly affects cell proliferation.

Have the authors waited longer with the models before starting treatment with the inhibitor? In other words, would the tumors regress if treatment were begun on day 8 instead of day 0 for the 143b model?

We apologize, if our treatment scheme shown in Fig. 7D was misleading. We did not start EGFR inhibitor treatment immediately after cell injection into the tibia bone, but did start the inhibitor treatment one week after the cells were injected and the tumors were small but already visible. It would not be feasible to wait another 8 days (=15 days after cell injection) before starting the treatment due to ethical reasons (the mice develop big tumors which results in paralysis of the whole leg).

7. Many of the legends to the Supplementary Figures are missing. This needs to be carefully checked in the text as well.

Minor points

P19 typo - "...were injected through.." not though

P 31 typo - Fig 6B legend "survival of patients" not patient.

P31- In the legend to Fig 7 panel B-C the figure is labeled cell viability (%). This should be labeled as an "increase in the number of viable cells", since a cell culture that is 100% viable cannot be made more viable- 150%.

P23 - In the author contribution section, EFW is not specifically mentioned.

Thank you for this information and apologies. We updated the missing parts of the Supplementary Figure legends and the other points in the revised manuscript.

Referee #3 (Comments on Novelty/Model System for Author):

Authors use an established genetically-engineered mouse model for induction of osteosarcoma with features of the human disease.

Referee #3 (Remarks for Author):

Osteosarcoma (OS) is an uncommon cancer affecting children and young adults. Metastatic osteosarcoma has a poor prognosis. Loss of p53 and Rb are common in OS, but the genetic driver landscape is otherwise complex. The authors have identified a role for Egfr in mouse bone development, and other reports describe high EGFR expression in a majority of human OS. A genetically engineered mouse model (H2-c-fosLTR) that induces high levels of c-fos induces OS. The authors report that endogenous egfr activity is important in activation of c-fos in this model, working through MAPK to CREB and Fos to induce OS. In vitro and in vivo inhibition or genetic loss of Egfr reduce OS formation and growth, concomitant with reduction in RSK2, Fos, and CREB signaling, reduced Cyclin D1, and reduced entry into cell cycle. This pathway is inhibited by MAPK inhibitor, but not PI3K pathway inhibitors. Over 200 weeks, high OS expression of FOS plus EGFR foreshadow poorer prognosis. EGFR-dependent cell line 143b but not EGF-insensitive line LM7, is growth-inhibited by erlotinib in xenografts. The authors conclude that EGFR inhibitors may be useful for control of a substantial subset of OS with high EGFR and FOS.

This is a clearly-written manuscript that attacks an important and challenging clinical problem. The data mainly well-controlled and convincing. The biological connection of EGFR to OS growth in the fosLTR model is interesting but not entirely surprising (although many other receptors could potentially have turned out to be involved.) An important finding is the connection of EGFR/FOS with prognosis in human OS in the context of the mouse model and xenograft results. The major biological issues are the relationship of the Fos model to human OS. More information on the OS biology in the context of these findings would enhance an already solid manuscript.

We would like to thank the reviewer for her/his very positive and constructive comments, which we have addressed as follows:

1. The authors recently reported the importance of EGFR in bone development. This means that the cellular landscape of bone and OS tissue in the egfr^{-/-} model may be very different from WT. In Fig.

3 and other related figures, does the loss of egfr and fos immunostaining reflect osteoblast cellularity or differentiation? Would this affect the frequency of cell-of-origin for FOS-driven OS, or, as the authors describe, is this mainly a signaling issue for activation of FOS.

Thank you for this very important question. For the revised version of the paper we looked into the cellular landscape and found that there is no apparent difference between c-fos transgenic OS from $Egfr^{DOB}$ and $Egfr^{wt}$ mice as shown by histological and mRNA expression analysis. We analyzed bone volume to total volume (BV/TV), as well as osteoclast number and osteocalcin protein and mRNA expression in the tumors and found no significant difference between the genotypes (Figs. EV2A-C). In addition, we also checked mRNA expression levels of other bone/osteoblast-specific differentiation marker that were partially deregulated in normal bones when EGFR was deleted, as recently published (Linder et al, 2018) and found no differences in mRNA isolated from OSs (Fig. EV2D). $Egfr^{DOB}$ mice develop a low-bone mass phenotype due to defects in osteoblast differentiation and proliferation. Therefore, we can formally not exclude that this also affects OS formation. However, mice harboring the hypomorphic EGFR waved-2 mutation do not develop a low-bone mass phenotype but still showed reduced OS formation and progression (Figs. 1A-D), suggesting that the bone phenotype of $Egfr^{DOB}$ mice does not affect c-Fos dependent OS formation.

Moreover, we did not see any significant differences in bone specific markers in OSs from H2-c-fosLTR / $Egfr^{DOB}$ vs H2-c-fosLTR / $Egfr^{wt}$ mice (Figs. EV2A-D), indicating that EGFR does not affect the cellularity in OSs but is rather essential for tumor development by directly activating FOS. This is also supported by our in vitro experiments, where we found a direct effect of EGFR signaling on c-Fos mRNA and protein expression as well as on proliferation/survival of isolated OS cells (Fig. 4).

2. Similarly, do double EGFR+ FOS+ human OS have different features (e.g. transcriptional profiles, putative genetic or epigenetic drivers) than the double negatives? This information may be available in public datasets.

We performed a LIMMA (Linear Models for Microarray Data and RNA-seq data) analysis of a publicly available dataset (E-GEOD-39058) and found no significant different expression profile between EGFR and FOS double high vs. double negative human OS (see Dataset EV1, page 12, line 263-266).

3. There is a great range of expression of EGFR in the six OS lines tested. Is high EGFR in 143b cell and in human OS associated with gene amplification? What are candidate ligands for EGFR in human OS?

The EGFR gene in 143b OS cells was sequenced in the OncoMap 3.0 of the CCLE and was found not to be mutated. Moreover, Wen et al also performed sequencing of exons 18 to 21 of the EGFR gene in the 143b cell line and found not functional mutation (Wen et al, 2007). This information is now also included in the paper (see page 20, line 434-435). We checked candidate ligands for EGFR in human OS by analyzing an available RNAseq dataset and found that Hb-EGF was significantly upregulated, as compared to other EGFR ligands (Fig. EV1F).

4. The partial inhibition of 143b xenografts by EGFR TKI is clear, but this level of response, with continued tumor growth, is not very strong for a xenograft model, and may suggest that EGFR inhibitors would not work well as single agents.

As we did not inject the OS cells subcutaneously but into the tibia bone, we think that this is a consequence of the orthotopic injection site which provides a more supportive tumor microenvironment where the cells growth better / faster. Nevertheless, for future clinical trials we would propose to use EGFR inhibition in combination with standard of care chemotherapy on OS patients with high FOS and EGFR.

Minor issues.

1. The blots and IHC are mostly excellent, but specificity is an issue of concern for IHC. How were human EGFR and FOS and other immunostaining validated?

Mouse IHC stainings were evaluated in an unbiased way using an IHC quantification program (imageJ with the plugin "Immunoratio"). Human EGFR and FOS IHC stainings were analyzed by two independent operators and results were in parallel double-checked with the histology program "definiens tissue studio 4.0".

2.No positive control to show that rapamycin and GSK agent work.

Rapamycin inhibits mTOR signaling, therefore we used pS6, a mTOR downstream signaling protein, as a control. GSK2334470 inhibits PDK1, we used pAKT which is downstream of PDK1, as a control. Both inhibitors worked as we did not see any pAKT and/or pS6 activation after EGF stimulation whereas DMSO pretreatment induced phosphorylation of both proteins (see Fig 4F).

3."EGFR addiction can be harnessed therapeutically to longlasting stable disease" may be over-optimistic. In EGFR lung adenocarcinoma, EGFR inhibitors have yet to control disease beyond 24 or so months on average time to progression, and worse for other diseases.

Thank you for raising this point. We meant that EGFR inhibition, in addition to standard of care treatment, could lead to long-lasting stable disease. We removed this half sentence in the revised manuscript.

References

David JP, Mehic D, Bakiri L, Schilling AF, Mandic V, Priemel M, Idarraga MH, Reschke MO, Hoffmann O, Amling M et al (2005) Essential role of RSK2 in c-Fos-dependent osteosarcoma development. *The Journal of clinical investigation* 115: 664-672

Linder M, Hecking M, Glitzner E, Zwerina K, Holcman M, Bakiri L, Ruocco MG, Tuckermann J, Schett G, Wagner EF et al (2018) EGFR controls bone development by negatively regulating mTOR-signaling during osteoblast differentiation. *Cell Death Differ* 25: 1094-1106

Walkley CR, Qudsi R, Sankaran VG, Perry JA, Gostissa M, Roth SI, Rodda SJ, Snay E, Dunning P, Fahey FH et al (2008) Conditional mouse osteosarcoma, dependent on p53 loss and potentiated by loss of Rb, mimics the human disease. *Genes Dev* 22: 1662-1676

Wen YH, Koeppen H, Garcia R, Chiriboga L, Tarlow BD, Peters BA, Eigenbrot C, Yee H, Steiner G, Greco MA (2007) Epidermal growth factor receptor in osteosarcoma: expression and mutational analysis. *Hum Pathol* 38: 1184-1191

2nd Editorial Decision

20 September 2018

Thank you for the submission of your revised manuscript to EMBO Molecular Medicine. We have now received the enclosed report from the referees that were asked to re-assess it. As you will see the reviewers are now supportive, and I am pleased to inform you that we will be able to accept your manuscript pending minor editorial amendments.

***** Reviewer's comments *****

Referee #1 (Remarks for Author):

I am satisfied with the responses of the authors to my comments

Referee #2 (Remarks for Author):

The authors did a very good job on the revised manuscript. I have no further comments.

Referee #3 (Comments on Novelty/Model System for Author):

The manuscript integrates mouse models and human data to identify EGFR-dependent regulation of FOS as a possible therapeutic vulnerability in human osteosarcoma.

Referee #3 (Remarks for Author):

The authors have responded to earlier critiques with new information and by clarifying the text.

Corresponding Author Name: Maria Sibilia
 Journal Submitted to: EMBO Molecular Medicine
 Manuscript Number: EMM-2018-09408